# A Simple Interpretable Transformer for Fine-Grained Image Classification and Analysis

**Dipanjyoti Paul**[1] **Arpita Chowdhury**[1] **Xinqi Xiong**[1] **Feng-Ju Chang**[2] **David Carlyn**[1]
**Samuel Stevens**[1] **Kaiya L. Provost**[1] **Anuj Karpatne**[3] **Bryan Carstens**[1] **Daniel Rubenstein**[4]
**Charles Stewart**[5] **Tanya Berger-Wolf**[1] **Yu Su**[1] **Wei-Lun Chao**[1]

[1]The Ohio State University  [2]Amazon Alexa  [3]Virginia Tech
[4]Princeton University  [5]Rensselaer Polytechnic Institute

**Painted Bunting!!** Do you see yourself? How do you interpret your decision?

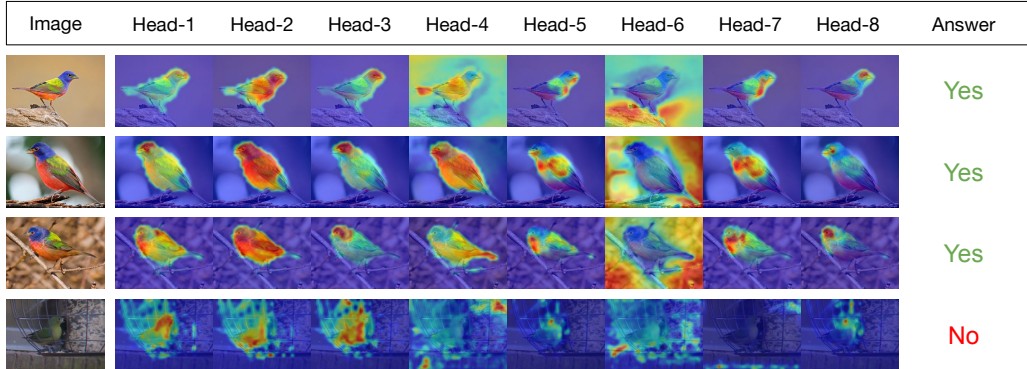

Figure 1: **Illustration of INTR**. We show four images (row-wise) of the same bird species Painted Bunting and the eight-head cross-attention maps (column-wise) triggered by the query of the ground-truth class. Each head is learned to attend to a different (across columns) but consistent (across rows) semantic cue in the image that is useful to recognize this bird species (e.g., attributes). The exception is the last row, which shows inconsistent attention. Indeed, this is a misclassified case, showcasing how INTR interprets (wrong) predictions.

## Abstract

We present a novel usage of Transformers to make image classification interpretable. Unlike mainstream classifiers that wait until the last fully connected layer to incorporate class information to make predictions, we investigate a *proactive* approach, asking each class to search for itself in an image. We realize this idea via a Transformer encoder-decoder inspired by DEtection TRansformer (DETR). We learn "class-specific" queries (one for each class) as input to the decoder, enabling each class to localize its patterns in an image via cross-attention. We name our approach INterpretable TRansformer (INTR), which is fairly easy to implement and exhibits several compelling properties. We show that INTR intrinsically encourages each class to attend distinctively; the cross-attention weights thus provide a faithful interpretation of the prediction. Interestingly, via "multi-head" cross-attention, INTR could identify different "attributes" of a class, making it particularly suitable for fine-grained classification and analysis, which we demonstrate on eight datasets. Our code and pre-trained models are publicly accessible at the Imageomics Institute GitHub site: `https://github.com/Imageomics/INTR`.

## 1 Introduction

Mainstream neural networks for image classification (He et al., 2016; Simonyan & Zisserman, 2015; Krizhevsky et al., 2017; Huang et al., 2019; Szegedy et al., 2015; Dosovitskiy et al., 2021; Liu et al., 2021) typically allocate most of their model capacity to extract "class-agnostic" feature vectors from

images, followed by a fully connected layer that compares image feature vectors with "class-specific" vectors to make predictions. While these models have achieved groundbreaking accuracy, their model design cannot directly explain *where* a model looks for predicting a particular class.

In this paper, we investigate a *proactive* approach to classification, asking each class to look for itself in an image. We hypothesize that this "class-specific" search process would reveal where the model looks, offering a built-in interpretation of the prediction.

At first glance, implementing this idea may need a significant model architecture design and a complex training process. However, we show that a novel usage of the Transformer encoder-decoder (Vaswani et al., 2017) inspired by DEtection TRansformer (DETR) (Carion et al., 2020) can essentially realize this idea, making our model fairly easy to reproduce and extend.

Concretely, the DETR encoder extracts patch-wise features from the image, and the decoder attends to them based on learnable queries. We propose to learn "class-specific" queries (one for each class) as input to the decoder, enabling the model to obtain "class-specific" image features via self-attention and cross-attention — self-attention encodes the contextual information among candidate classes, determining the patterns necessary to distinguish between classes; cross-attention then allows each class to look for the distinctive patterns in the image. The resulting "class-specific" image feature vectors (one for each class) are then compared with a shared "class-agnostic" vector to predict the label of the image. We name our model INterpretable TRansformer (INTR). Figure 2 illustrates the model architecture. In the training phase, we learn INTR by minimizing the cross-entropy loss. In the inference phase, INTR allows us to visualize the cross-attention maps triggered by different "class-specific" queries to understand why the model predicts or does not predict a particular class.

On the surface, INTR may fall into the debate of whether attention is interpretable (Jain & Wallace, 2019; Wiegreffe & Pinter, 2019; Bibal et al., 2022). However, we mathematically show that INTR offers faithful attention to distinguish between classes. In short, INTR computes logits by performing inner products between class-specific feature vectors and the shared class-agnostic vector. To classify an image correctly, the ground-truth class must obtain distinctive class-specific image features to claim the highest logit against other classes, which is possible only through distinct cross-attention weights. Minimizing the training loss thus encourages each class-specific query to produce distinct cross-attention weights. Manipulating the cross-attention weights in inference, as done in adversarial attacks to attention-based interpretation (Serrano & Smith, 2019), would alter the prediction notably.

We extensively analyze INTR, especially in cross-attention. We find that the "multiple heads" in cross-attention could learn to identify different "attributes" of a class and consistently localize them in images, making INTR particularly well-suited for fine-grained classification. We validate this on multiple datasets, including CUB-200-2011 (Wah et al., 2011), Birds-525 (Piosenka, 2023), Oxford Pet (Parkhi et al., 2012), Stanford Dogs (Khosla et al., 2011), Stanford Cars (Krause et al., 2013), FGVC-Aircraft (Maji et al., 2013), iNaturalist-2021 (Van Horn et al., 2021), and Cambridge butterfly (Montejo-Kovacevich et al., 2020). Interestingly, by concentrating the decoder's input on visually similar classes (e.g., the mimicry in butterflies), INTR could attend to the nuances of patterns, even matching those found by biologists, suggesting its potential benefits to scientific discovery.

It is worth reiterating that INTR is built upon a widely-used Transformer encoder-decoder architecture and can be easily trained end-to-end. *What makes it interpretable is the novel usage — incorporating class-specific information at the decoder's input rather than output.* We view these as key strengths and contributions. They make INTR easily applicable, reproducible, and extendable.

## 2 BACKGROUND AND RELATED WORK

### 2.1 WHAT KIND OF INTERPRETATIONS ARE WE LOOKING FOR?

As surveyed in (Zhang & Zhu, 2018; Burkart & Huber, 2021; Carvalho et al., 2019; Das & Rad, 2020; Buhrmester et al., 2021; Linardatos et al., 2020), various ways exist to explain or interpret a model's prediction (see Appendix A for more details). Among them, the most popular is localizing *where* the model looks for predicting a particular class. We follow this notion and focus on fine-grained classification (e.g., bird and butterfly species). That is, not only do we want to localize the coarse-grained objects (e.g., birds and butterflies), but we also want to identify the "attributes" (e.g., wing patterns) that are useful to distinguish between fine-grained classes. We note that an attribute can

be decomposed into "object part" (e.g., head, tail, wing, etc.) and "property" (e.g., patterns on the wings), in which the former is commonly shared across all classes (Wah et al., 2011). We thus expect that our approach could identify the differences within a part between classes, not just localize parts.

## 2.2 BACKGROUND AND NOTATION

We denote an image and its ground-truth label by $I$ and $y$, respectively. To perform classification over $C$ classes, mainstream neural networks learn a feature extractor $f_{\boldsymbol{\theta}}$ to obtain a feature map $\boldsymbol{X} = f_{\boldsymbol{\theta}}(\boldsymbol{I}) \in \mathbb{R}^{D \times H \times W}$. Here, $\boldsymbol{\theta}$ denotes the parameters; $D$ denotes the number of channels; $H$ and $W$ denote the number of grids in the height and width dimensions. For instance, ResNet (He et al., 2016) realizes $f_{\boldsymbol{\theta}}$ by a convolutional neural network (ConvNet) with residual links; Vision Transformer (ViT) (Dosovitskiy et al., 2021) realizes it by a Transformer encoder. Normally, this feature map is reshaped and/or pooled into a feature vector denoted by $\boldsymbol{x} = \text{Vect}(\boldsymbol{X})$, which then undergoes inner products with $C$ class-specific vectors $\{\boldsymbol{w}_{\text{c}}\}_{c=1}^{C}$. The class with the largest inner product is outputted as the predicted label,

$$\hat{y} = \arg\max_{c \in [C]} \quad \boldsymbol{w}_{\text{c}}^{\top} \boldsymbol{x}. \tag{1}$$

## 2.3 RELATED WORK ON POST-HOC EXPLANATION AND SELF-INTERPRETABLE METHODS

Since this classification process does not explicitly localize *where* the model looks to make predictions, the model is often considered a black box. To explain the prediction, a post-hoc mechanism is needed (Ribeiro et al., 2016; Koh & Liang, 2017; Yuan et al., 2021; Qiang et al., 2022; Zhou et al., 2015). For instance, CAM (Zhou et al., 2016) and Grad-CAM (Selvaraju et al., 2017) obtain class activation maps (CAM) by back-propagating class-specific gradients to the feature map. RISE (Petsiuk et al., 2018) iteratively masks out image contents to identify essential regions for classification. These methods have been widely used. However, they are often low-resolution (e.g., blurred or indistinguishable across classes), computation-heavy, and not necessarily aligned with how models make predictions.

To address these drawbacks, another branch of work designs models with interpretable prediction processes, incorporating explicit mechanisms that allow for a direct understanding of the predictions (Wang et al., 2021; Donnelly et al., 2022; Rigotti et al., 2021; Kim et al., 2022; Bau et al., 2017; Zhou et al., 2018). For example, ProtoPNet (Chen et al., 2019) compares the feature map $\boldsymbol{X}$ to "learnable prototypes" of each class, resulting in a feature vector $\boldsymbol{x}$ whose elements are semantically meaningful: the $d$-th dimension corresponds to a prototypical part of a certain class and $x[d]$ indicates its activation in the image. By reading $\boldsymbol{x}$ and visualizing the activated prototypes, one could better understand the model's decision. Inspired by ProtoPNet, ProtoTree (Nauta et al., 2021) arranges the comparison to prototypes in a tree structure to mimic human reasoning; ProtoPFormer (Xue et al., 2022) presents a Transformer-based realization of ProtoPNet, which was originally based on ConvNets. Along with these interpretable decision processes, however, come specifically tailored architecture designs and increased complexity of the training process, often making them hard to reproduce, adapt, or extend. For instance, ProtoPNet requires a multi-stage training strategy, each stage taking care of a portion of the learnable parameters including the prototypes.

# 3 INTERPRETABLE TRANSFORMER (INTR)

## 3.1 MOTIVATION AND BIG PICTURE

Taking into account the pros and cons of the above two paradigms, we ask, *Can we obtain interpretability via standard neural network architectures and standard learning algorithms?*

To respond to "*interpretability*", we investigate a *proactive* approach to classification, asking each class to search for its presence and distinctive patterns in an image. Denote by $\mathcal{S}$ the set of candidate classes; we propose a new classification rule,

$$\hat{y} = \arg\max_{c \in [C]} \quad \boldsymbol{w}^{\top} g_{\boldsymbol{\phi}}(f_{\boldsymbol{\theta}}(\boldsymbol{I}), c, \mathcal{S}), \tag{2}$$

where $g_{\boldsymbol{\phi}}(f_{\boldsymbol{\theta}}(\boldsymbol{I}), c, \mathcal{S})$ represents the image feature vector extracted specifically for class $c$ in the context of $\mathcal{S}$, and $\boldsymbol{w}$ denotes a binary classifier determining whether class $c$ is present in the image $\boldsymbol{I}$. Compared to Equation 1, the new classification rule in Equation 2 incorporates class-specific

information in the feature extraction stage, not in the final fully connected layer. As will be shown in subsection 3.4, this design is the key to generating faithful attention for interpretation.

To respond to "*standard neural network architectures*", we find that the Transformer encoder-decoder (Vaswani et al., 2017), which is widely used in object detection (Carion et al., 2020; Zhu et al., 2021) and natural language processing (Wolf et al., 2020), could essentially realize Equation 2. Specifically, the encoder extracts the image feature map $X = f_{\boldsymbol{\theta}}(\boldsymbol{I})$. For the decoder, we propose to learn $C$ class-specific queries $\{z_{\text{in}}^{(c)}\}_{c=1}^{C}$ as input, enabling it to extract the feature vector $g_{\boldsymbol{\phi}}(f_{\boldsymbol{\theta}}(\boldsymbol{I}), z_{\text{in}}^{(c)}, \mathcal{S})$ for class $c$ via cross-attention.

To ease the description, let us first focus on cross-attention, the key building block in Transformer decoders in subsection 3.2. We then introduce our full model in subsection 3.3.

## 3.2 INTERPRETABLE CLASSIFICATION VIA CROSS-ATTENTION

**Cross-attention.** Cross-attention can be seen as a (soft) retrieval process. Given an input query vector $z_{\text{in}} \in \mathbb{R}^D$, it finds similar vectors from a vector pool and combines them via weighted average. In our application, this pool corresponds to the feature map $X$. Without loss of generality, let us reshape the feature map $X \in \mathbb{R}^{D \times H \times W}$ to $X = [x_1, \cdots, x_N] \in \mathbb{R}^{D \times N}$. That is, $X$ contains $N = H \times W$ feature vectors representing each spatial grid in an image; each vector $x_n$ is $D$-dimensional.

With $z_{\text{in}}$ and $X$, cross-attention performs the following sequence of operations. First, it projects $z_{\text{in}}$ and $X$ to a common embedding space such that they can be compared, and separately projects $X$ to another space to emphasize the information to be combined,

$$q = W_{\text{q}} z_{\text{in}} \in \mathbb{R}^D, \quad K = W_{\text{k}} X \in \mathbb{R}^{D \times N}, \quad V = W_{\text{v}} X \in \mathbb{R}^{D \times N}. \quad (3)$$

Then, it performs an inner product between $q$ and $K$, followed by Softmax, to compute the similarities between $z_{\text{in}}$ and vectors in $X$, and uses the similarities as weights to combine vectors in $V$ linearly,

$$z_{\text{out}} = V \times \mathsf{Softmax}(\frac{K^{\top} q}{\sqrt{D}}) \in \mathbb{R}^D, \quad (4)$$

where $\sqrt{D}$ is a scaling factor based on the dimensionality of features. In other words, the output of cross-attention is a vector $z_{\text{out}}$ that aggregates information in $X$ according to the input query $z_{\text{in}}$.

**Class-specific queries.** Inspired by the inner workings of cross-attention, we propose to learn $C$ "class-specific" query vectors $Z_{\text{in}} = [z_{\text{in}}^{(1)}, \cdots, z_{\text{in}}^{(C)}] \in \mathbb{R}^{D \times C}$, one for each class. We expect each of these queries to look for the "class-specific" distinctive patterns in $X$. The output vectors $Z_{\text{out}} = [z_{\text{out}}^{(1)}, \cdots, z_{\text{out}}^{(C)}] \in \mathbb{R}^{D \times C}$ thus should encode whether each class finds itself in the image,

$$Z_{\text{out}} = V \times \mathsf{Softmax}(\frac{K^{\top} Q}{\sqrt{D}}) \in \mathbb{R}^{D \times C}, \quad \text{where } Q = W_{\text{q}} Z_{\text{in}} \in \mathbb{R}^{D \times C}. \quad (5)$$

We note that the Softmax is taken over elements of each column; i.e., in Equation 5, each column in $Z_{\text{in}}$ attends to $X$ independently. We use superscript/subscript to index columns in $Z/X$.

**Classification rule.** We compare each vector in $Z_{\text{out}}$ to a learnable "presence" vector $w \in \mathbb{R}^D$ to determine whether each class is found in the image. The predicted class is thus

$$\hat{y} = \arg\max_{c \in [C]} \quad w^{\top} z_{\text{out}}^{(c)}. \quad (6)$$

**Training.** As each class obtains a logit $w^{\top} z_{\text{out}}^{(c)}$, we employ the cross-entropy loss,

$$\ell(\boldsymbol{I}, y) = -\log \frac{\exp(w^{\top} z_{\text{out}}^{(y)})}{\sum_{c'} \exp(w^{\top} z_{\text{out}}^{(c')})}, \quad (7)$$

coupled with stochastic gradient descent (SGD) to optimize the learnable parameters, including $Z_{\text{in}}$, $w$, and the projection matrices $W_{\text{q}}$, $W_{\text{k}}$, and $W_{\text{v}}$ in Equation 3. This design responds to the final piece of question in subsection 3.1, "*standard learning algorithms*".

**Inference and interpretation.** We follow Equation 6 to make predictions. Meanwhile, each column of the cross-attention weights $\mathsf{Softmax}(\frac{K^{\top} Q}{\sqrt{D}})$ in Equation 5 reveals where each class looks to find

itself, enabling us to understand why the model predicts or does not predict a class. We note that this built-in interpretation does not incur additional computation costs like post-hoc explanation.

**Multi-head attention.** It is worth noting that a standard cross-attention block has multiple heads. It learns multiple sets of matrices $(\boldsymbol{W}_{\text{q, r}}, \boldsymbol{W}_{\text{k, r}}, \boldsymbol{W}_{\text{v, r}})$ in Equation 3, $r \in \{1, \cdots, R\}$, to look for different patterns in $\boldsymbol{X}$, resulting in multiple $\mathsf{Softmax}(\frac{\boldsymbol{K}_r^\top \boldsymbol{Q}_r}{\sqrt{D}})$ and $\boldsymbol{Z}_{\text{out, r}}$ in Equation 5. *This enables the model to identify different "attributes" of a class and allows us to visualize them.*

In training and inference, $\{\boldsymbol{Z}_{\text{out, r}}\}_{r=1}^R$ are concatenated row-wise, followed by another learnable matrix $\boldsymbol{W}_{\text{o}}$ to obtain a single $\boldsymbol{Z}_{\text{out}}$ as in Equation 5,

$$\boldsymbol{Z}_{\text{out}} = \boldsymbol{W}_{\text{o}}[\boldsymbol{Z}_{\text{out, 1}}^\top, \cdots, \boldsymbol{Z}_{\text{out, R}}^\top]^\top \in \mathbb{R}^D. \tag{8}$$

As such, Equation 7 and Equation 6 are still applicable to optimize the model and make predictions.

## 3.3 Overall model architecture (see Figure 2 for an illustration)

We implement our full INterpretable TRansformer (INTR) model (cf. Equation 2) using a Transformer decoder (Vaswani et al., 2017) on top of a feature extractor $f_{\boldsymbol{\theta}}$ that produces a feature map $\boldsymbol{X}$. Without loss of generality, we use the DEtection TRansformer (DETR) (Carion et al., 2020) as the backbone. DETR uses a Transformer decoder of multiple layers; each contains a cross-attention block. The output vectors of one layer become the input vectors of the next layer. In DETR, the input to the decoder (at its first layer) is a set of object proposal queries, and we replace it with our learnable "class-specific" query vectors $\boldsymbol{Z}_{\text{in}} = [\boldsymbol{z}_{\text{in}}^{(1)}, \cdots, \boldsymbol{z}_{\text{in}}^{(C)}] \in \mathbb{R}^{D \times C}$. The Transformer decoder then outputs the "class-specific" feature vectors $\boldsymbol{Z}_{\text{out}}$ that will be fed into Equation 6.

Using a Transformer decoder rather than a single cross-attention block has several advantages. First, with multiple decoder layers, the learned queries $\boldsymbol{Z}_{\text{in}}$ can improve over layers by grounding themselves on the image. Second, the self-attention block in each decoder layer allows class-specific queries to exchange information to encode the context. (See Appendix C for details.) As shown in Figure 15, the cross-attention blocks in later layers can attend to more distinctive patterns.

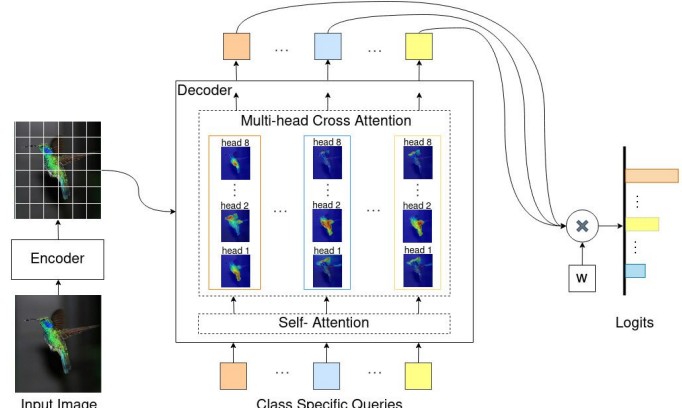

Figure 2: Model architecture of **INTR**. See subsection 3.2 for details.

**Training.** INTR has three sets of learnable parameters: a) the parameters in the DETR backbone, including $f_{\boldsymbol{\theta}}$; b) the class-specific input queries $\boldsymbol{Z}_{\text{in}} \in \mathbb{R}^{D \times C}$ to the decoder; and c) the class-agnostic vector $\boldsymbol{w}$. We train all these parameters end-to-end via SGD, using the loss in Equation 7.

## 3.4 How does INTR learn to produce interpretable cross-attention weights?

We analyze how INTR offers interpretability. For brevity, we focus on the model in subsection 3.2.

**Attention vs. interpretation.** There has been an ongoing debate on whether attention offers faithful interpretation (Wiegreffe & Pinter, 2019; Jain & Wallace, 2019; Serrano & Smith, 2019; Bibal et al., 2022). Specifically, Serrano & Smith (2019) showed that significantly manipulating the attention weights at inference time does not necessarily change the model's prediction. Here, we provide a mathematical explanation for why INTR may not suffer from the same problem. The key is in our classification rule. In Equation 6, we obtain the logit for class $c$ by $\boldsymbol{w}^\top \boldsymbol{z}_{\text{out}}^{(c)}$. If $c$ is the ground-truth label, it must obtain a logit *larger* than other classes $c' \neq c$ to make a correct prediction. This implies $\boldsymbol{z}_{\text{out}}^{(c)} \neq \boldsymbol{z}_{\text{out}}^{(c')}$, which is possible only if the cross-attention weights triggered by $\boldsymbol{z}_{\text{in}}^{(c)}$ are different from those triggered by other class-specific queries $\boldsymbol{z}_{\text{in}}^{(c')}$ (cf. Equation 4 and Equation 5 for how $\boldsymbol{z}_{\text{out}}^{(c)}$ is constructed). Minimizing the training loss in Equation 7 thus would force each learnable query vector $\boldsymbol{z}_{\text{in}}^{(c)}, \forall c \in [C]$, to be distinctive and able to attend to class-specific patterns in the input image.

**Unveiling the inner workings.** We dig deeper to understand what INTR learns. For class $c$ to obtain a high logit in Equation 6, $z_{\text{out}}^{(c)}$ must have a large inner product with the class-agnostic $w$. We note that $z_{\text{out}}^{(c)}$ is a linear combination of $V = [v_1, \cdots, v_N] \in \mathbb{R}^{D \times N}$ (cf. Equation 4), and $V$ is obtained by applying a projection matrix $W_{\text{v}}$ to the feature map $X = [x_1, \cdots, x_N] \in \mathbb{R}^{D \times N}$ (cf. Equation 3). Let $q^{(c)} = W_{\text{q}} z_{\text{in}}^{(c)}$ and let $\alpha^{(c)} = \text{Softmax}(\frac{K^\top q^{(c)}}{\sqrt{D}}) \in \mathbb{R}^N$, the logit $w^\top z_{\text{out}}^{(c)}$ can be rewritten as

$$
\begin{aligned}
w^\top z_{\text{out}}^{(c)} = w^\top V \alpha^{(c)} &\propto [w^\top v_1, \cdots, w^\top v_N] \exp(K^\top q^{(c)}) \\
&\propto [w^\top W_{\text{v}} x_1, \cdots, w^\top W_{\text{v}} x_N] \exp\left((W_{\text{k}} X)^\top q^{(c)}\right) \\
&\propto [s_1, \cdots, s_N] \exp([q^{(c)\top} W_{\text{k}} x_1, \cdots, q^{(c)\top} W_{\text{k}} x_N]^\top) = \sum_n s_n \times \alpha^{(c)}[n],
\end{aligned}
\tag{9}
$$

where $s_n = w^\top W_{\text{v}} x_n$ and $\alpha^{(c)}[n] \propto \exp(q^{(c)\top} W_{\text{k}} x_n)$. We note that $s_n$ does not depend on the class-specific query $z_{\text{in}}^{(c)}$. It only depends on the input image $I$, or more specifically, the feature map $X$ and how it aligns with the vector $w$. In other words, we can view $s_n$ as an "image-specific" salient score for patch $n$. In contrast, $\alpha^{(c)}[n]$ depends on the class-specific query $z_{\text{in}}^{(c)}$; its value will be high if class $c$ finds the distinctive patterns in patch $n$.

Building on this insight and Equation 9, if class $c$ is the ground-truth class, what its query $z_{\text{in}}^{(c)}$ needs to do is putting its attention weights $\alpha^{(c)}$ on those high-score patches. Namely, class $c$ must find its distinctive patterns in the salient image regions. Putting things together, we can view the roles of $W_{\text{v}}$ and $W_{\text{k}}$ as "disentanglement". They disentangle the information in $x_n$ into "image-specific" and "classification-specific" components — the former highlights "whether a patch should be looked at"; the latter highlights "what distinctive patterns it contains". When multi-head cross-attention is used, each pair of $(W_{\text{v}}, W_{\text{k}})$ can learn to highlight an object "part" and the distinctive "property" in it. These offer the opportunity to localize the "attributes" of a class. See Appendix B for more details.

### 3.5 COMPARISON TO CLOSELY RELATED WORK

**ProtoPNet (Chen et al., 2019) and Concept Transformers (CT) (Rigotti et al., 2021).** INTR is fundamentally different in two aspects. First, both methods aim to represent image patches by a set of learnable vectors (e.g., prototypes in ProtoPNet; concepts in CT [1]). The resulting features for image patches are then pooled into a vector $x$ and undergo a fully connected layer for classification. In other words, their classification rules still follow Equation 1. In contrast, INTR extracts class-specific features from the image (one per class) and uses a new classification rule to make predictions (cf. Equation 6). Second, both methods require specifically designed training strategies or signals. For example, CT needs human annotations to learn the concepts. In contrast, INTR is based on a standard model architecture and training algorithm and requires no additional human supervision.

**DINO-v1 (Caron et al., 2021).** DINO-v1 shows that the "[CLS]" token of a pre-trained ViT (Dosovitskiy et al., 2021) can attend to different "parts" of objects via multi-head attention. While this shares some similarities with our findings in INTR, what INTR attends to are "attributes" that can be used to distinguish between fine-grained classes, not just "parts" that are shared among classes.

## 4 EXPERIMENTS

**Dataset.** We consider eight fine-grained datasets from various domains, including Birds-525 (**Bird**) (Piosenka, 2023), CUB-200-2011 Birds (**CUB**) (Wah et al., 2011),

Table 1: Dataset statistics. (# images are rounded.)

|  | Bird | CUB | BF | Fish | Dog | Pet | Car | Craft |
|---|---|---|---|---|---|---|---|---|
| # Train Img. | 85K | 6K | 5K | 45K | 12K | 4K | 8K | 3K |
| # Test Img. | 3K | 6K | 1K | 2K | 9K | 4K | 8K | 3K |
| # Classes | 525 | 200 | 65 | 183 | 120 | 37 | 196 | 100 |

Cambridge butterfly (**BF**) (Montejo-Kovacevich et al., 2020), iNaturalist-2021-Fish (**Fish**) (Van Horn et al., 2021), Stanford Dogs (**Dog**) (Khosla et al., 2011), Stanford Cars (**Car**) (Krause et al., 2013), Oxford Pet (**Pet**) (Parkhi et al., 2012), and FGVC Aircraft (**Craft**) (Maji et al., 2013). We create the BF dataset by considering the species-level labels from (Montejo-Kovacevich et al., 2020). For the Fish dataset, we extract species from the taxonomical *Class* named *Animalia Chordata Actinopterygii* in iNaturalist-2021. Table 1 provides the dataset statistics. See Appendix D for additional details.

---

[1] Even though CT applies cross-attention, it uses image patches as queries to attend to the concept embeddings; the outputs of cross-attention are thus features for image patches.

**Model.** We implement INTR on top of the DETR backbone (Carion et al., 2020). DETR stacks a Transformer encoder on top of a ResNet as the feature extractor. We use its DETR-ResNet-50 version, in which the ResNet-50 (He et al., 2016) was pre-trained on ImageNet-1K (Russakovsky et al., 2015; Deng et al., 2009) and the whole model including the Transformer encoder-decoder (Vaswani et al., 2017) was further trained on MSCOCO (Lin et al., 2014)[2]. We remove its prediction heads located on top of the decoder and add our class-agnostic vector $w$; we remove its object proposal queries and add our $C$ learnable class-specific queries (e.g., for CUB, $C = 200$). See Figure 2 for an illustration and subsection 3.3 for more details. We further remove the positional encoding that was injected into the cross-attention keys in the DETR decoder: we find this information adversely restricts our queries to look at particular grid locations and leads to artifacts. We note that DETR sets its feature map size $D \times H \times W$ (at the encoder output) as $256 \times \frac{H_0}{32} \times \frac{W_0}{32}$, where $H_0$ and $W_0$ are the height and width resolutions of the input image. For example, a typical CUB image is of a resolution roughly $800 \times 1200$; thus, the resolution of the feature map and cross-attention map is roughly $25 \times 38$. We investigate other encoders and the number of attention heads and decoder layers in Appendix F.

**Visualization.** We visualize the **last** (i.e., **sixth**) decoder layer, whose cross-attention block has **eight heads**. We superimpose the cross-attention weight (maps) on the input images.

**Training detail.** The hyper-parameter details such as epochs, learning rate, and batch size for training INTR are reported in Appendix E. We use the Adam optimizer (Kingma & Ba, 2014) with its default hyper-parameters. We train INTR using the StepLR scheduler with a learning rate drop at 80 epochs. The rest of the hyper-parameters follow DETR.

**Baseline.** We consider two sets of baseline methods. First, we use a ResNet-50 (He et al., 2016) pre-trained on ImageNet-1K and fine-tune it on each dataset. We then use Grad-CAM (Selvaraju et al., 2017) and RISE Petsiuk et al. (2018) to construct post-hoc saliency maps: the results are kept in Appendix F. Second, we compare to models designed for interpretability, such as ProtoPNet (Chen et al., 2019), ProtoTree (Nauta et al., 2021), and ProtoPFormer (Xue et al., 2022). We understand that these are by no means a comprehensive set of existing works. Our purpose in including them is to treat them as references for what kind of interpretability INTR can offer with its simple design.

**Evaluation.** *We reiterate that achieving a high classification accuracy is not the goal of this paper. The goal is to demonstrate the interpretability.* We thus focus our evaluation on qualitative results.

## 4.1 EXPERIMENTAL RESULTS

Table 2: Accuracy(%) comparison.

| Model | Bird | CUB | BF | Fish | Dog | Pet | Car | Craft |
|---|---|---|---|---|---|---|---|---|
| ResNet | 98.5 | 83.8 | 95.6 | 71.9 | 77.1 | 89.5 | 89.3 | 80.9 |
| INTR | 97.4 | 71.8 | 95.0 | 81.1 | 72.5 | 90.4 | 86.8 | 76.1 |

**Accuracy comparison.** It is crucial to emphasize that the primary objective of INTR is to promote interpretability, not to claim high accuracy. Nevertheless, we report in Table 2 the classification accuracy of INTR and ResNet-50 on all eight datasets. INTR obtains comparable accuracy on most of the datasets except for CUB (12% worse) and Fish (9.2% better). We note that both CUB and Bird datasets focus on fine-grained bird species. The main difference is that the Bird dataset offers higher-quality images (e.g., cropped to focus on objects). INTR's accuracy drop on CUB thus more likely results from its inability to handle images with complex backgrounds or small objects, not its inability to recognize bird species.

**Comparison to interpretable models.** We compare INTR to ProtoPNet, ProtoTree, and ProtoPFormer (Figure 3). For the compared methods, we show the responses of the top three prototypes (sorted by their activations in the image) of the ground-truth class. For INTR, we show the top three cross-attention maps (sorted by the peak un-normalized attention weight in the map) triggered by the ground-true class. INTR can identify distinctive attributes similarly to the other methods. In particular, INTR is capable of localizing tiny attributes (like patterns of beaks and eyes): unlike the other methods, INTR does not need to pre-define the patch size of a prototype or attribute.

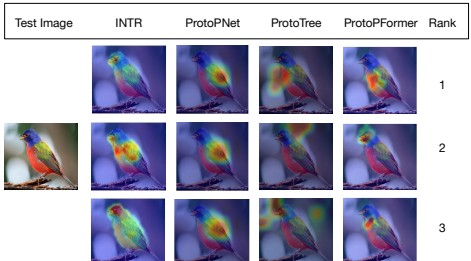

Figure 3: Comparison to interpretable models. We show the responses of the top three cross-attention heads or prototypes (row-wise) of each method (column-wise) in a Painted Bunting image.

---

[2]Please see subsection 4.2 for a discussion on concerns about data leakage and unfair comparison.

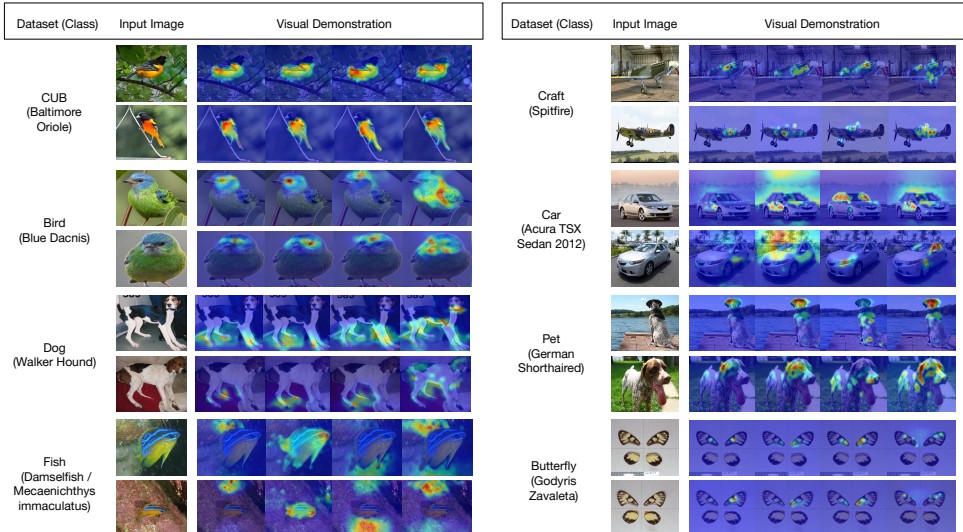

Figure 4: **INTR on all eight datasets.** We show the **top four** cross-attention maps per test example triggered by the ground-truth classes (based on the peak un-normalized attention weights in the maps). As the indices of the top maps may not be the same across test examples, the attributes may not be the same in each column.

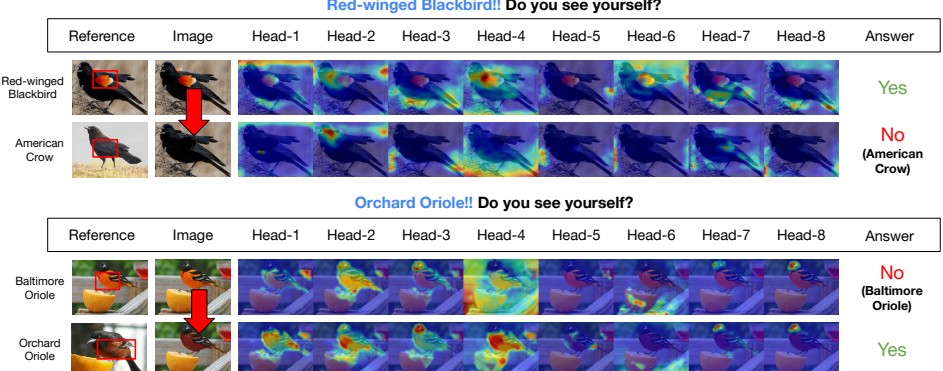

Figure 5: **INTR can identify tiny image manipulations that distinguish between classes.** On the top, we remove the red spots of the Red-winged Blackbird. After that, INTR cannot correctly classify the image — the parentheses in the Answer column highlight the predicted classes. On the bottom, we change the color of the bird's belly (Baltimore Oriole) to make it look like Orchard Oriole. After that, INTR would misclassify it as Orchard Oriole. Both results demonstrate INTR's sensitivity to visual attributes.

## 4.2 FURTHER ANALYSIS AND DISCUSSION ABOUT INTR

**INTR can consistently identify attributes.** We first analyze whether different cross-attention heads identify different attributes of a class and if those attributes are consistent across images of the same class. Figure 1 shows a result (please see the caption for details). Different columns correspond to different heads, and we see that each captures a distinct attribute that is consistent across images. Some of them are very fine-grained, such as Head-4 (tail pattern) and Head-5 (breast color). The reader may notice the less concentrated attention in the last row. Indeed, it is a misclassified case: the query of the ground-truth class (i.e., Painted Bunting) cannot find itself in the image. This showcases how INTR interprets incorrect predictions. We show more results in Appendix G.

**INTR is applicable to a variety of domains.** Figure 4 shows the cross-attention results on all eight datasets. (See the caption for details.) INTR can identify the attributes well in all of them, demonstrating its remarkable generalizability and applicability.

**INTR offers meaningful interpretation about attribute manipulation.** We investigate INTR's response to image manipulation by deleting (the first block of Figure 5) and adding (the second block of Figure 5) important attributes. We obtain human-identified attributes of *Red-winged Blackbird* (the first block) and *Orchard Oriole* (the second block) from (Cor) and manipulate them accordingly.

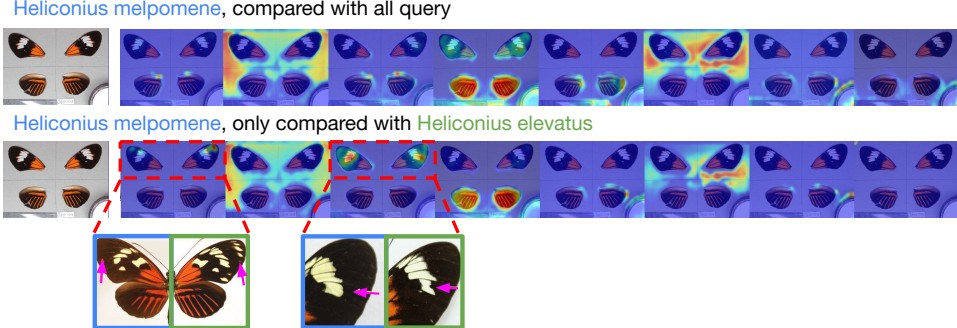

Figure 6: **INTR can identify fine-grained attributes to differentiate visually similar classes.** The test image (first column) is Heliconius melpomene. We show the cross-attention maps triggered by the ground-truth class when compared to all other classes (first row) and the visually similar Heliconius elevatus (second row). As shown in the second row, limiting the input queries to visually similar classes enables INTR to identify the nuances of patterns, even matching those found by biologists. Specifically, at the bottom, we show the image of Heliconius melpomene (blue box) and Heliconius elevatus (green box) and where biologists localize the attributes (purple arrows). The bottom images are taken from (Hel).

As shown in Figure 5, INTR is sensitive to the attribute changes; the cross-attention maps change drastically at the manipulated parts. These results suggest that INTR's inner working is heavily dependent on attributes to make correct classifications.

**INTR can attend differently based on the context.** As mentioned in subsection 3.3, the self-attention block in INTR's decoder could encode the context of candidate classes to determine the patterns necessary to distinguish between them. When all the class-specific queries (e.g., 65 classes in the BF dataset) are inputted to the decoder, INTR needs to identify sufficient patterns (e.g., both coarse-grained and fine-grained) to distinguish between all of them. Here, we investigate whether limiting the input queries to visually similar ones would encourage the model to attend to finer-grained attributes. We focus on the BF dataset and compare two species, Heliconius melpomene (blue box in Figure 6) and Heliconius elevatus (green box in Figure 6), whose visual difference is very subtle. We limit the input queries by setting other queries as zero vectors. As shown in Figure 6, this modification does allow INTR to localize nuances of patterns between the two classes.

**Concerns regarding an MSCOCO-pre-trained backbone.** We understand this may cause concern about data leakage and unfair comparison. We note that MSCOCO only offers bounding boxes for objects, not for parts, and it does not contain fine-grained labels. Regarding fair comparisons, our work is not to claim higher accuracy but to offer a new perspective. We use DETR to demonstrate that our idea can be easily compatible with pre-trained encoder-decoder (foundation) models.

**Limitations.** INTR learns $C$ class-specific queries that must be inputted to the Transformer decoder *jointly*. This could increase the training and inference time if $C$ is huge, e.g., larger than the number of grids $N$ in the feature map. Fortunately, fine-grained classification (e.g., for species in the same family or order) usually focuses on a small set of visually similar categories; $C$ is usually not large.

## 5 CONCLUSION

We present Interpretable Transformer (INTR), a simple yet effective interpretable classifier building upon standard Transformer encoder-decoder architectures. INTR makes merely two changes: learning class-specific queries (one for each class) as input to the decoder and learning a class-agnostic vector on top of the decoder output to determine whether a class is present in the image. As such, INTR can be easily trained end-to-end. During inference, the cross-attention weights triggered by the winning class-specific query indicate where the model looks to make the prediction. We conduct extensive experiments and analyses to demonstrate the effectiveness of INTR in interpretation. Specifically, we show that INTR can localize not only object parts like bird heads but also attributes (like patterns around eyes) that distinguish one bird species from others. In addition, we present a mathematical explanation of why INTR can learn to produce interpretable cross-attention for each class without ad-hoc model design, complex training strategies, and auxiliary supervision. We hope that our study can offer a new way of thinking about interpretable machine learning.

## ACKNOWLEDGMENT

This research is supported in part by grants from the National Science Foundation (IIS-2107077 and OAC-2118240). We are thankful for the generous support of the computational resources by the Ohio Supercomputer Center. We thank Lisa Wu (OSU) for a fruitful discussion on datasets.

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

# APPENDIX

We provide details omitted in the main paper.

- Appendix A: related work (cf. subsection 2.1 of the main paper).
- Appendix B: additional details of inner workings and visualization (cf. subsection 3.2 and subsection 3.4 of the main paper).
- Appendix C: additional details of model architectures (cf. subsection 3.3 of the main paper).
- Appendix D: details of dataset (cf. section 4 of the main paper).
- Appendix E: details of experimental setup (cf. section 4 of the main paper).
- Appendix F: additional experimental results (subsection 4.1 of the main paper).
- Appendix G: additional qualitative results and analysis (cf. subsection 4.1 of the main paper).
- Appendix H: additional discussion (cf. section 5 of the main paper).

## A RELATED WORK

In recent years, there has been a significant increase in the size and complexity of models, prompting a surge in research and development efforts focused on enhancing model interpretability. The need for interpretability arises not only from the goal of instilling trust in a model's predictions but also from the desire to comprehend the reasoning behind a model's predictions, gain insight into its internal mechanisms, and identify the specific input features it relies on to make accurate predictions. Numerous research directions have emerged to facilitate model interpretation for human understanding. One notable research direction involves extracting and visualizing the salient regions in an input image that contribute to the model's prediction. By identifying these regions, researchers aim to provide meaningful explanations that highlight the relevant aspects of the input that influenced the model's decision. Existing efforts in this domain can be broadly categorized into post hoc methods and self-interpretable models.

Post hoc methods involve applying interpretation techniques after a model has been trained. These methods focus on analyzing the model's behavior without modifying its architecture or training process. Most CNN-based classification processes lack explicit information on where the model focuses its attention during prediction. Post hoc methods address this limitation by providing interpretability and explanations for pre-trained black box models without modifying the model itself. For instance, CAM (Zhou et al., 2016) computes a weighted sum of feature maps from the last convolutional layer based on learned fully connected layer weights, generating a single heat map highlighting relevant regions for the predicted class. GRAD-CAM (Selvaraju et al., 2017) employs gradient information flowing into the last convolutional layer to produce a heatmap, with the gradients serving as importance weights for feature maps, emphasizing regions with the greatest impact on the prediction. Koh & Liang (2017) introduce influence functions, which analyze gradients of the model's loss function with respect to training data points, providing a measure of their influence on predictions. Another approach in post hoc methods involves perturbing or sampling the input image. For example, LIME (Ribeiro et al., 2016) utilizes superpixels to generate perturbations of the input image and explain predictions of a black box model. RISE (Petsiuk et al., 2018) iteratively blocks out parts of the input image, classifies the perturbed image using a pre-trained model, and reveals the blocked regions that lead to misclassification. However, post hoc methods for model interpretation can be computationally expensive, making them less scalable for real-world applications. Moreover, these methods may not provide precise explanations or a comprehensive understanding of how the model makes decisions, affecting the reliability and robustness of the interpretation results obtained.

Self-interpretable models are designed with interpretability as a core principle. These models incorporate explicit mechanisms or structures that allow for a direct understanding of their decision-making process. One direction is prototype-based models. Prototypes are visual representations of concepts that can be used to explain how a model works. The first work of using prototypes to describe the DNN model's prediction is ProtoPNet (Chen et al., 2019), which learns a predetermined number of prototypical parts (prototypes) per class. To classify an image, the model calculates the

similarity between a prototype and a patch in the image. This similarity is measured by the distance between the two patches in latent space. Inspired by ProtoPNet, ProtoTree (Nauta et al., 2021) is a hierarchical neural network architecture that learns class-agnostic prototypes approximated by a decision tree. This significantly decreases the required number of prototypes for interpreting a prediction than ProtoPNet.

ProtoPNet and its variants were originally designed to work with CNN-based backbones. However, they can also be used with ViTs (Vision Transformer) by removing the class token. This approach, however, has several limitations. First, prototypes are more likely to activate in the background than in the foreground. When activated in the foreground, their activation is often scattered and fragmented. Second, prototype-based methods are computationally heavy and require domain knowledge to fix the parameters. With the widespread use of transformers in computer vision, many approaches have been proposed to interpret their classification predictions. These methods often rely on attention weights to visualize the important regions in the image that contribute to the prediction. ProtoPFormer addresses this problem by applying the prototype-based method to ViTs. However, these prototype-based methods are computationally expensive and require domain knowledge to set the parameters. ProtoPFormer (Xue et al., 2022) works on solving the problem by applying the prototype-based method with ViTs. However, these prototype-based works are computationally heavy and require domain knowledge to fix the parameters. ViT-Net (Kim et al., 2022) integrates ViTs and trainable neural trees based on ProtoTree, which only uses ViTs as feature extractors without fully exploiting their architectural characteristics. Another recent work, Concept Transformer (Rigotti et al., 2021), utilizes patch embeddings of an image as queries and attributes from the dataset as keys and values within a transformer. This approach allows the model to obtain multi-head attention weights, which are then used to interpret the model's predictions. However, a drawback of this method is that it relies on human-defined attribute annotations for the dataset, which can be prone to errors and is costly as it necessitates domain expert involvement.

## B    ADDITIONAL DETAILS OF INNER WORKINGS AND VISUALIZATION

**Interpretability vs. model capacity.** We investigate whether the conventional classification rule in Equation 1 induces the same property discussed in subsection 3.4. We replace $\boldsymbol{w}^\top \boldsymbol{z}_{\text{out}}^{(c)}$ with $\boldsymbol{w}_c^\top \boldsymbol{z}_{\text{out}}^{(c)}$; i.e., we learn for each class a class-specific $\boldsymbol{w}_c$. This can be thought of as increasing the model capacity by introducing additional learnable parameters. The resulting classification rule is

$$\hat{y} = \arg\max_{c \in [C]} \quad \boldsymbol{w}_c^\top \boldsymbol{z}_{\text{out}}^{(c)}. \tag{10}$$

Here, even if $\boldsymbol{z}_{\text{out}}^{(c)} = \boldsymbol{z}_{\text{out}}^{(c')}$, class $c$ can still claim the highest logit as long as $\boldsymbol{z}_{\text{out}}^{(c)}$ has a larger inner product with $\boldsymbol{w}_c$ than other $\boldsymbol{w}_{c'}^\top \boldsymbol{z}_{\text{out}}^{(c')}$. Namely, even if the cross-attention weights triggered by different class-specific queries are identical,[3] as long as the extracted features in $\boldsymbol{X}$ are correlated strongly enough with class $c$, the model can still predict correctly. Thus, the learnable queries $\boldsymbol{z}_{\text{in}}^{(c)}, \forall c \in [C]$, need not necessarily learn to produce distinct and meaningful cross-attention weights.

Indeed, as shown in Figure 7, we implement a variant of our approach INTR-FC with its classification rule replaced by Equation 10. INTR produces more distinctive (column-wise) and consistent (row-wise) attention.

**Visualization.** In subsection 3.4 of the main paper, we show how the logit of class $c$ can be decomposed into

$$\boldsymbol{w}^\top \boldsymbol{z}_{\text{out}}^{(c)} = \sum_n s_n \times \alpha^{(c)}[n],$$

$$\text{where} \quad s_n = \boldsymbol{w}^\top \boldsymbol{W}_{\text{v}} \boldsymbol{x}_n; \tag{11}$$

$$\boldsymbol{\alpha}^{(c)}[n] \propto \exp(\boldsymbol{q}^{(c)^\top} \boldsymbol{W}_{\text{k}} \boldsymbol{x}_n) = \exp(\boldsymbol{z}_{\text{in}}^{(c)^\top} \boldsymbol{W}_{\text{q}}^\top \boldsymbol{W}_{\text{k}} \boldsymbol{x}_n).$$

The index $n$ corresponds to a grid location (or column) in the feature map $\boldsymbol{X} \in \mathbb{R}^{D \times N}$.

---

[3]In the extreme case, one may consider the weights to be uniform, i.e., $\frac{1}{N}$, at all spatial grids for all classes.

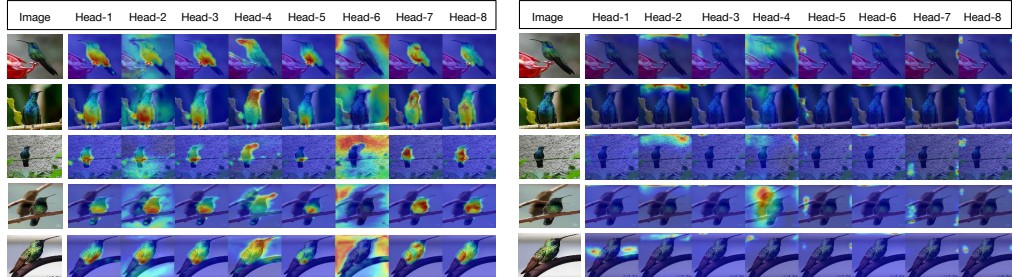

Figure 7: INTR (left columns) vs. INTR-FC (right columns). INTR produces better interpretations than INTR-FC. The bird species is Green Violetear.

Based on Equation 11, to predict an input image as class $c$, the cross-attention map $\boldsymbol{\alpha}^{(c)}$ triggered by the class-specific query $\boldsymbol{z}_{\text{in}}^{(c)}$ should align with the image-specific scores $[s_1, \cdots, s_N]$. In other words, for an image that is predicted as class $c$, the cross-attention map $\boldsymbol{\alpha}^{(c)}$ very much implies which grids in an image have higher scores. Hence, in the qualitative visualizations, we only show the cross-attention map $\boldsymbol{\alpha}^{(c)}$ rather than the image-specific scores.

We note that throughout the whole paper, INTR learns to identify attributes that are useful to distinguish classes *without* relying on the knowledge of human experts.

## C  ADDITIONAL DETAILS OF MODEL ARCHITECTURES

Our idea in subsection 3.2 can be realized by the standard Transformer decoder (Vaswani et al., 2017) on top of any "class-agnostic" feature extractors that produce a feature map $\boldsymbol{X}$ (e.g., ResNet (He et al., 2016) or ViT (Dosovitskiy et al., 2021)). A Transformer decoder often stacks $M$ layers of the same decoder architecture denoted by $\{L_m\}_{m=1}^M$. Each layer $L_m$ takes a set of $C$ vector tokens as input and produces another set of $C$ vector tokens as output, which can then be used as the input to the subsequent layer $L_{m+1}$. In our application, the learnable "class-specific" query vectors $\boldsymbol{Z}_{\text{in}} = [\boldsymbol{z}_{\text{in}}^{(1)}, \cdots, \boldsymbol{z}_{\text{in}}^{(C)}] \in \mathbb{R}^{D \times C}$ are the input tokens to $L_1$.

Within each decoder layer is a sequence of building blocks. Without loss of generality, let us omit the layer normalization, residual connection, and Multi-Layer Perceptron (MLP) operating on each token independently, but focus on the Self-Attention (SA) and the subsequent Cross-Attention (CA) blocks.

An SA block is very similar to the CA block introduced in subsection 3.1. The only difference is the pool of vectors to be retrieved — while a CA block attends to the feature map extracted from the image, *the SA block attends to its input tokens*. That is, in an SA block, the $\boldsymbol{X} \in \mathbb{R}^{D \times N}$ matrix in Equation 3 is replaced by the input matrix $\boldsymbol{Z}_{\text{in}} \in \mathbb{R}^{D \times C}$. This allows each query token $\boldsymbol{z}_{\text{in}}^{(c)} \in \mathbb{R}^D$ to combine information from other query tokens, resulting in a new set of $C$ query tokens. This new set of query tokens is then fed into a CA block that attends to the image features in $\boldsymbol{X}$ to generate the "class-specific" feature tokens.

As a Transformer decoder stacks multiple layers, the input tokens to the second layers and beyond possess not only the "learnable" class-specific information in $\boldsymbol{Z}_{\text{in}}$ but also the class-specific feature information from $\boldsymbol{X}$. We note that an SA block can aggregate information not only from similar tokens[4] but also from dissimilar tokens. For example, when $\boldsymbol{W}_{\text{q}}$ is an identity matrix and $\boldsymbol{W}_{\text{k}} = -\boldsymbol{W}_{\text{q}}$, a pair of similar tokens in $\boldsymbol{Z}_{\text{in}}$ will receive smaller weights than a pair of dissimilar tokens. This allows similar query tokens to be differentiated if their relationships to other tokens are different, enabling the model to distinguish between semantically or visually similar fine-grained classes.

---

[4]In this paragraph, this refers to the similarity in the inner product space.

Table 3: Statistics of the datasets from different domains.

|          | Bird  | CUB  | BF   | Fish  | Dog   | Pet  | Car  | Craft |
|----------|-------|------|------|-------|-------|------|------|-------|
| Training | 84635 | 5994 | 4697 | 45162 | 12000 | 3680 | 8144 | 3334  |
| Testing  | 2625  | 5794 | 275  | 1830  | 8580  | 3669 | 8041 | 3333  |
| Classes  | 525   | 200  | 65   | 183   | 120   | 37   | 196  | 100   |

## D    DETAILS OF DATASETS

We present the detailed dataset statistics in  Table 3. We download the butterfly (**BF**) dataset from the Heliconiine Butterfly Collection Records[5] at the University of Cambridge. These downloaded datasets exhibit class imbalances. To address this, we performed a selection process on the downloaded data as follows: First, we consider classes with a minimum of $B$ images, where $B$ is set to 20. Subsequently, for each class, we retained at least $K$ images for testing, with $K$ set to 3. Throughout this process, we also ensured that we had no more than $M$ training images, where $M$ is defined as 5 times the quantity $(B - K)$. The resulting dataset statistics are presented in  Table 3.

## E    DETAILS OF EXPERIMENTAL SETUP

During our experiment, for all datasets, except for Bird, we set the learning rate to $1 \times e^{-4}$, while for Bird, we use a learning rate of $5 \times e^{-5}$. Additionally, we utilize a batch size of 16 for Bird, Dog, and Fish datasets, and a batch size of 12 for the other datasets. Furthermore, the number of epochs required for training is 100 for BF and Pet datasets, 170 for Dog, and 140 for the remaining datasets.

## F    ADDITIONAL EXPERIMENTAL RESULTS

**Performance with different encoder backbones.** In subsection 4.1, we demonstrate that INTR yields consistent results when compared to ResNet50. We further delve into an analysis of INTR's performance using various architectural configurations, specifically by employing different encoder backbones. Our objective is to ascertain whether INTR can potentially achieve superior performance with an alternative encoder backbone. To investigate this, we employ DeiT (Touvron et al., 2021) and ViT (Dosovitskiy et al., 2021) models pre-trained on ImageNet-1K and ImageNet-21K datasets, respectively. Specifically, we utilize DeiT-Small (INTR-DeiT-S-1K) and ViT-Huge (INTR-ViT-H-21K). The results of our investigation are presented in Table 4: we see that using different encoders can impact the accuracy. Specifically, on CUB where we see a huge drop of INTR in Table 2, using a ViT-Huge backbone can achieve a $11.4\%$ improvement.

Table 4: Performance of INTR using different encoders.

| Encoder Backbone | CUB  | BF   | Car  |
|------------------|------|------|------|
| INTR (DETR-R50)  | 71.8 | 95.0 | 86.8 |
| INTR (DeiT-S-1K) | 78.2 | 94.2 | 81.8 |
| INTR (ViT-H-21K) | 83.2 | 95.7 | 88.4 |

Table 5: INTR with different numbers of decoder layers and attention heads, using the CUB dataset.

| Algorithm | Number of heads | | | Number of decoders | | | |
|-----------|-------|-------|-------|-------|-------|-------|-------|
|           | 4     | 8     | 16    | 4     | 5     | 6     | 7     |
| INTR      | 69.48 | **71.75** | 70.17 | 68.39 | 69.21 | **71.75** | 69.07 |

We further perform ablation studies on different numbers of attention heads and decoder layers. The results are reported in Table 5. We find that the setup by DETR (i.e., 8 heads and 6 decoder layers) performs the best.

---

[5]https://zenodo.org/record/3477412

**Comparisons to post-hoc explanation methods.** We use Grad-CAM (Selvaraju et al., 2017) and RISE (Petsiuk et al., 2018) to construct post-hoc saliency maps on the ResNet-50 (He et al., 2016) classifiers. We also report the insertion and deletion metric scores (Petsiuk et al., 2018) to quantify the results. It is worth mentioning that insertion and deletion metrics were designed to quantify post-hoc explanation methods. However, here we show a comparison between INTR, RISE, and Grad-CAM. We examine the CUB dataset images that are accurately classified by both ResNet50 and INTR, resulting in

Table 6: Quantitative comparisons of interpretations utilizing ResNet as a common backbone on CUB dataset. For insertion, the higher, the better. For deletion, the lower, the better.

| Algorithm | Insertion | Deletion |
|---|---|---|
| GradCam | 0.61 | 0.31 |
| RISE | 0.56 | 0.30 |
| INTR | **0.786** | **0.01** |

a reduced count of $3,582$ validation images. We generate saliency maps using Grad-CAM and INTR and then rank the patches to assess the insertion and deletion metrics. For a fair comparison, we employ ResNet-50 as a shared classifier for evaluation. The results are reported in Table 6.

We further compare the attention map of INTR (averaged over eight heads) to the saliency map of Grad-CAM and RISE (using ResNet-50). All methods can visualize the model's responses to different classes. For INTR, this is through the cross-attention triggered by different queries. In Figure 8, we show a correctly (left) and a wrongly (right) classified example. The two test images on the top are from the same class (i.e., Rufous Hummingbird), and we visualize the model's responses to four candidate classes (row-wise; exemplar images are provided for reference); the first row is the ground-truth class. Resolution-wise, both INTR and Grad-CAM show sharper saliency. Discrimination-wise, INTR clearly identifies where each class sees itself — each attention map highlights where the candidate class and the test image look alike. Such a message is not clear from Grad-CAM and RISE. Interestingly, in the case where both INTR and ResNet-50 predict wrongly (into the fourth class: Ruby-Throated Hummingbird), INTR is able to interpret the decision: the similarity between the test image and the fourth class seems more pronounced compared to the true class. Indeed, the notable attribute of the true class (i.e., the bright orange head) is not clearly shown in the test image in Figure 8 (b).

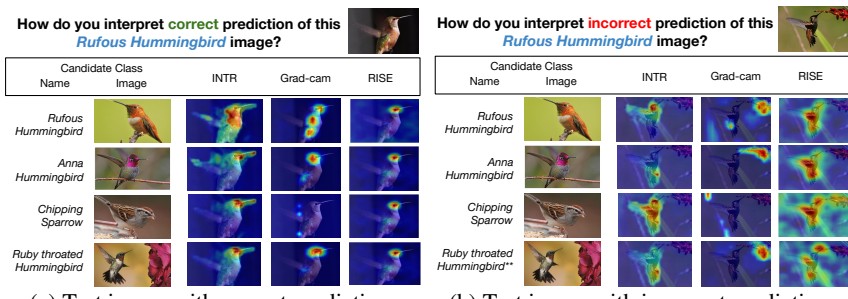

(a) Test image with correct prediction.  (b) Test image with incorrect prediction.

Figure 8: Comparison to Grad-CAM to RISE. The test image is at the top. Each row is a candidate class. The columns of INTR, Grad-CAM, and RISE show the model's response to each candidate class in the test image.

## G  ADDITIONAL QUALITATIVE RESULTS AND ANALYSIS

**Figure 9** offers additional results of Figure 3. In Figure 3 of the main paper, we visualize the top three cross-attention heads or prototypes. Figure 9 further shows all the prototypes or attention heads for the same test image featuring the Painted Bunting species.

To gain further insights into the detected attributes, we compare INTR with ProtoPFormer, a prominent method in our previous evaluations. We randomly picked five images from each of the four species sampled uniformly from the CUB dataset. Figure 10, shows the attention heads detected by these methods for four images, each from a different species. We validate the attributes detected by these methods through a human study. We provide detected attention heads and image-level attribute information (available in the CUB metadata) to seven individuals, who are unfamiliar with the work. We instruct them to list all attributes they believe are captured by the attention heads. An attribute is deemed detected if more than half of the individuals identify it from the attention

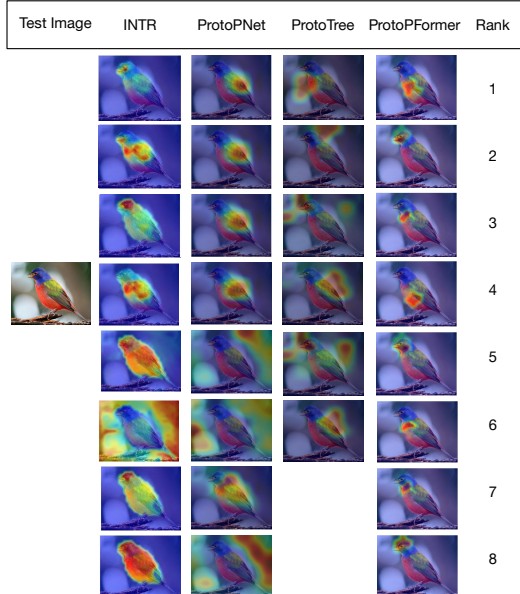

Figure 9: Extended Comparison with ProtoPNet, ProtoPFormer, and ProtoTree.

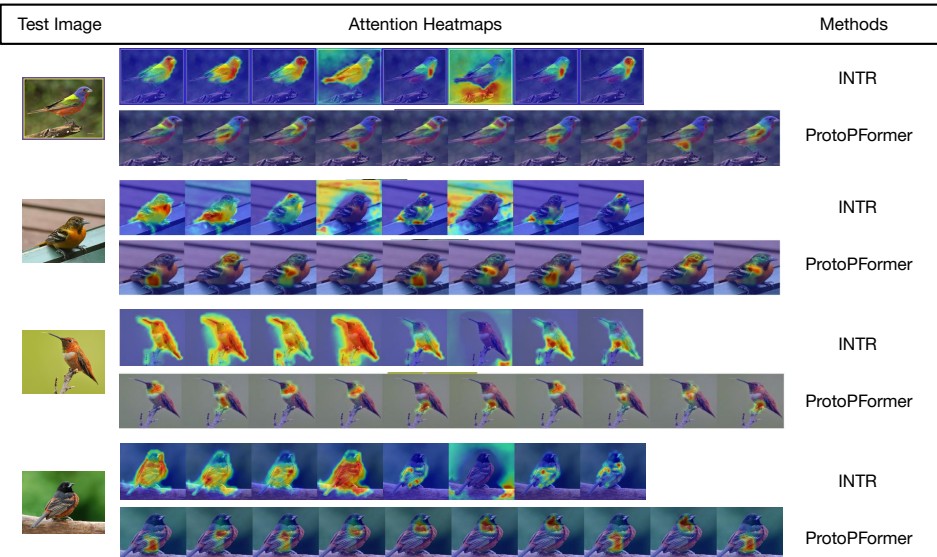

Figure 10: Extended Comparison between INTR, and ProtoPFormer. We show attention heads comparison between INTR, and ProtoPFormer for four images, each from a different species.

heads. Our quantitative analysis reveals that INTR outperforms ProtoPFormer, achieving an attribute identification accuracy of 74.7% compared to ProtoPFormer's 42.2%.

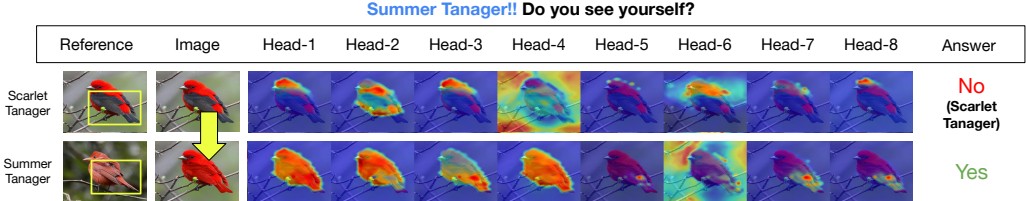

Figure 11: **INTR can identify tiny image manipulations that distinguish between classes.** We change the color of the Scarlet Tanager's wings and tail to make it look like Summer Tanager. After that, INTR would misclassify it as Summer Tanager. The result demonstrates INTR's sensitivity to visual attributes.

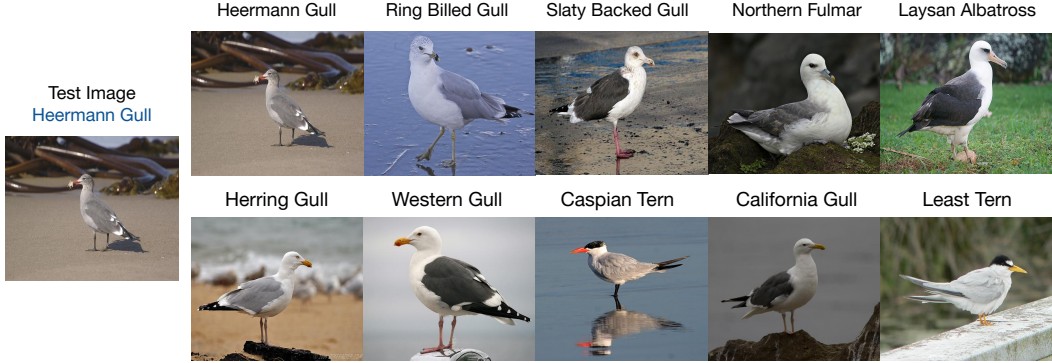

Figure 12: We present the top ten classes predicted by INTR for the test image in the Heermann Gull (CUB dataset), showcasing similarities in appearances and genera among them.

**Heermann Gull!! Do you see yourself? How do you interpret your decision?**

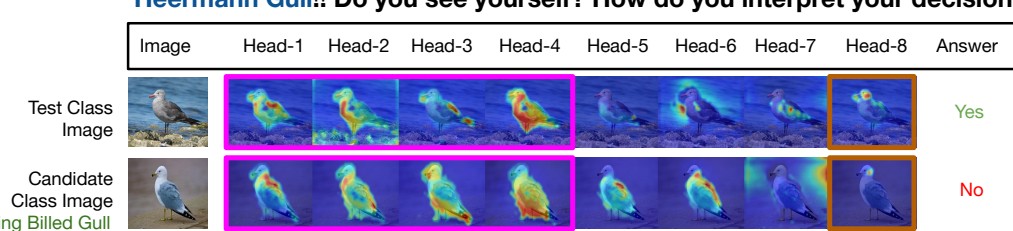

Figure 13: **INTR's Class-specific query can able to discriminate similar species.** The test image (first row) is Heermann Gull and the most similar candidate class (second row) is Ring Billed Gull. We show the cross-attention maps of the ground-truth class image (first row) and the candidate class image (second row) triggered by the test class ground-truth query. The query searches for class-specific attributes in both species. For instance, in Head-1 to Head-4 (purple box), both rows detect the common back, breast, tail, and belly pattern respectively. Head-8 (brown box) detects the red black-tipped bill from the test class but not the yellow ring bill from the candidate class.

In the main paper, we demonstrate the capability of INTR in detecting tiny image manipulations, focusing on the species Red-winged Blackbird and Orchard Oriole as detailed in Figure 5. We further extend our analysis to another species, Scarlet Tanager, in Figure 11. Specifically, we modified the Scarlet Tanager by altering its wing and tail colors to red, resembling the Summer Tanager. These alterations were conducted following the attribute guidelines (Cor). To quantitatively measure the effects, we randomly selected ten images from each species for manipulation. Our observations revealed that twenty-nine out of thirty cases resulted in a change in classification post-manipulation, indicating a success rate of 96.7%. This underscores INTR's ability to discern tiny image modifications that differentiate between distinct classes.

**How does INTR differentiate similar classes?** We explore the predictive capabilities and investigate INTR's ability to recognize classes that share similar attributes. In Figure 12, we show the top predicted classes by INTR for the test image Heermann Gull. The top ten predictions indeed exhibit similar appearances and genera, indicating the meaningfulness of these predictions.

We further investigate attributes detected by INTR that are responsible for distinguishing similar species. The attributes that INTR captures are local patterns (specific shape, color, or texture) useful to characterize a species or differentiate between species. These attributes can be shared across species if the species are visually similar. These can be seen in Figure 13 and Figure 14. In Figure 13, we applied the query of Heermann Gull to the image of Heermann Gull (first row) and Ring Billed Gull (second row). Since these two species are visually similar, several attention heads identify similar attributes from both images. However, at Head-8, the attention maps clearly identify the unique attribute of Heermann Gull that is not shown in Ring Billed Gull. Please see the caption for details. In Figure 14, we present the cross-attention map activated by the ground-truth queries for two closely related species, Baltimore Oriole and Orchard Oriole. Additionally, we manually document some of the attributes by checking whether the attention maps align with those human-annotated attributes in the CUB dataset. This reveals that INTR can identify shared and discriminative attributes in similar classes.

## Baltimore Oriole!! Do you see yourself? How do you interpret your decision?

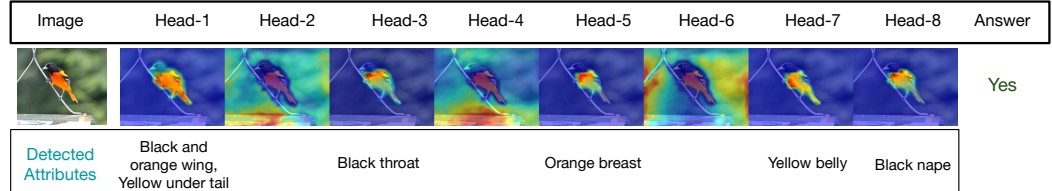

## Orchard Oriole!! Do you see yourself? How do you interpret your decision?

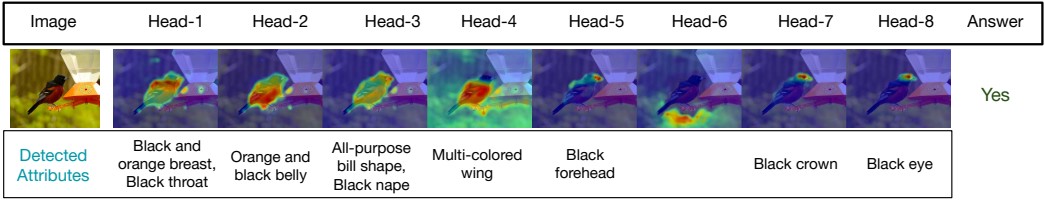

Figure 14: **INTR can identify shared and discriminative attributes in similar classes.** We present the cross-attention map activated by the query corresponding to the ground-truth class for two closely related species, the Baltimore Oriole and the Orchard Oriole. Additionally, we document the attributes identified by class-specific queries, as evaluated by humans. It is worth noting that certain attributes detected, such as black throat, breast color, etc., are shared, given the similarity of these two species.

**Class-specific queries are improved over decoder layers.** As mentioned in section 4 and Appendix B, our implementation of INTR has six decoder layers; each contains one cross-attention block. In qualitative results, we only show the cross-attention maps from the sixth layer, which produces the class-specific features that will be compared with the class-agnostic vector for prediction (cf. Equation 6). For the cross-attention blocks in other decoder layers, their output feature tokens become the input (query) tokens to the subsequent decoder layers. That is, the class-specific queries will change (and perhaps, improve) over layers.

To illustrate this, we visualize the cross-attention maps produced by each decoder layer. The results are in Figure 15. The attention maps improve over layers in terms of the attributes they identify so as to differentiate different classes.

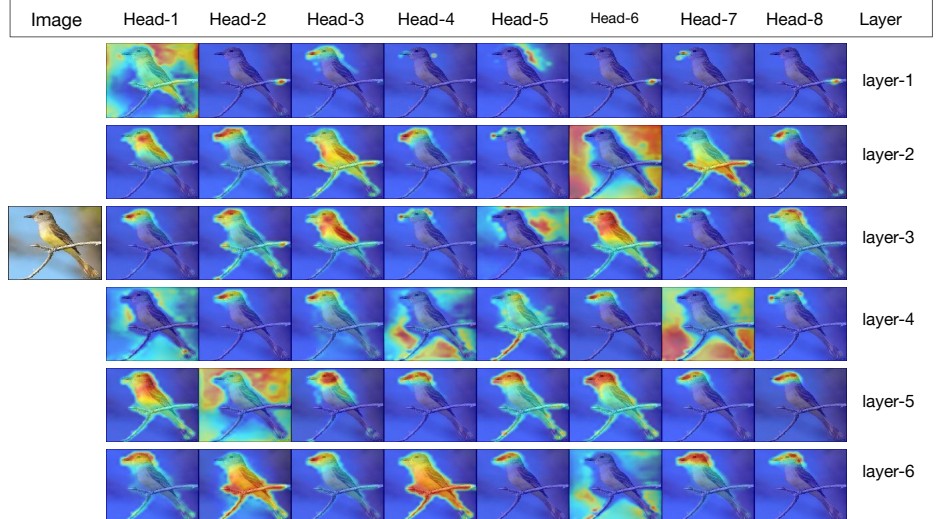

Figure 15: **Attention maps from different INTR Decoder layers**, across different cross-attention heads on the same image using the true query.

**Figure 16, Figure 17, Figure 18, Figure 19, Figure 20, Figure 21, and Figure 22** showcase the cross-attention maps associated with the dataset Bird, BF, Dog, Pet, Fish, Craft, and Car respectively, following the same format of Figure 1. Remarkably, different heads of INTR for each dataset successfully identify different attributes of classes, and these attributes remain consistent across images of the same class. INTR provides the interpretation of its prediction on different datasets across domains.

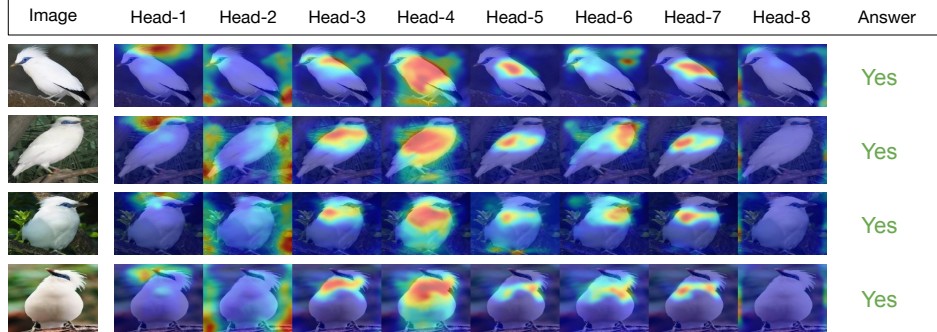

Figure 16: **Illustration of INTR**. We show four images (row-wise) of the same bird species Bali Starling and the eight-head cross-attention maps (column-wise) triggered by the query of the ground-truth class. Each head is learned to attend to a different (across columns) but consistent (across rows) semantic cue in the image that is useful to recognize this bird species.

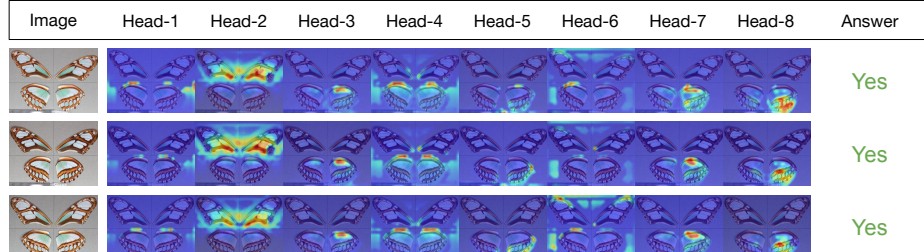

Figure 17: **Illustration of INTR**. We show three images (row-wise) of the same butterfly species Paititia Neglecta and the eight-head cross-attention maps (column-wise) triggered by the query of the ground-truth class. Each head is learned to attend to a different (across columns) but consistent (across rows) semantic cue in the image that is useful to recognize this butterfly species.

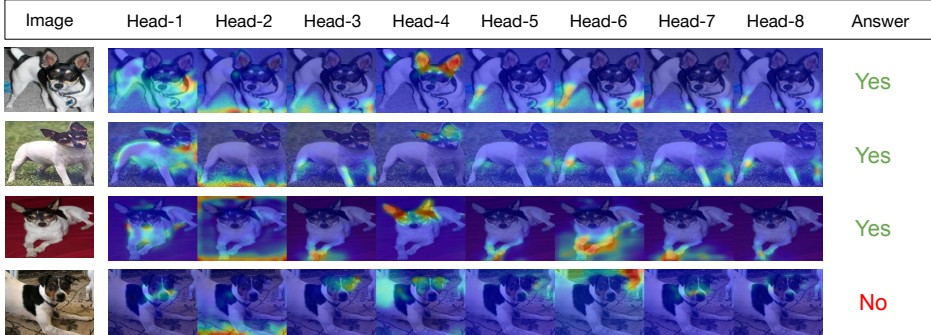

Figure 18: **Illustration of INTR**. We show four images (row-wise) of the same dog breed Toy Terrier and the eight-head cross-attention maps (column-wise) triggered by the query of the ground-truth class. Each head is learned to attend to a different (across columns) but consistent (across rows) semantic cue in the image that is useful to recognize this dog breed. The exception is the last row, which shows inconsistent attention. Indeed, this is a misclassified case, showcasing how INTR interprets (wrong) predictions.

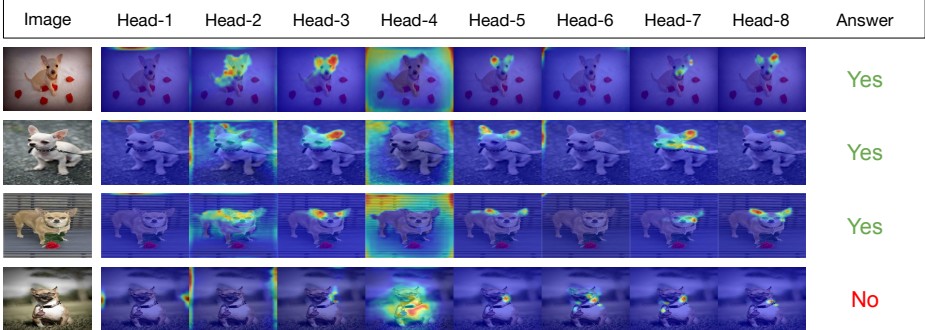

Figure 19: **Illustration of INTR**. We show four images (row-wise) of the same pet dog breed Chihuahua (from the Pet dataset) and the eight-head cross-attention maps (column-wise) triggered by the query of the ground-truth class. Each head is learned to attend to a different (across columns) but consistent (across rows) semantic cue in the image that is useful to recognize this pet breed. The exception is the last row, which shows inconsistent attention. Indeed, this is a misclassified case, showcasing how INTR interprets (wrong) predictions.

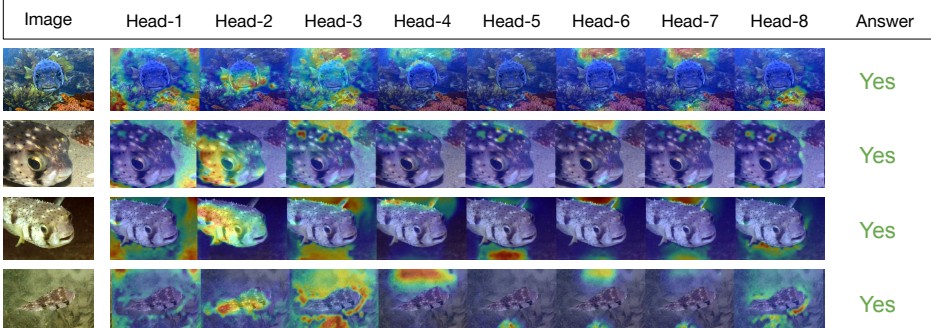

Figure 20: **Illustration of INTR**. We show four images (row-wise) of the same fish species Dicotylichthys punctulatus and the eight-head cross-attention maps (column-wise) triggered by the query of the ground-truth class. Each head is learned to attend to a different (across columns) but consistent (across rows) semantic cue in the image that is useful to recognize this fish species.

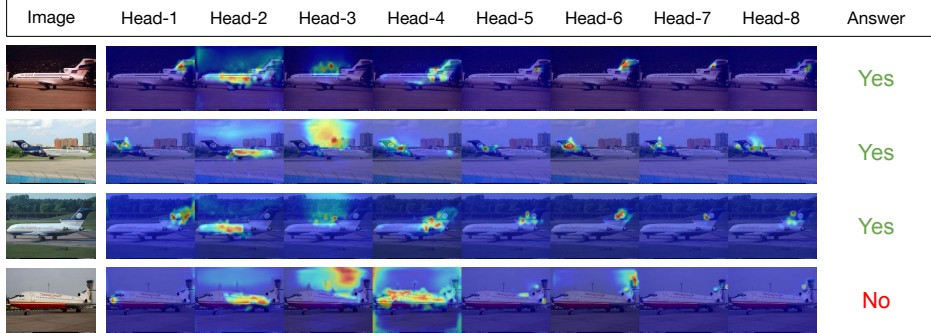

Figure 21: **Illustration of INTR**. We show four images (row-wise) of the same craft variant A319 and the eight-head cross-attention maps (column-wise) triggered by the query of the ground-truth class. Each head is learned to attend to a different (across columns) but consistent (across rows) semantic cue in the image that is useful to recognize this craft variant. The exception is the last row, which shows inconsistent attention. Indeed, this is a misclassified case, showcasing how INTR interprets (wrong) predictions.

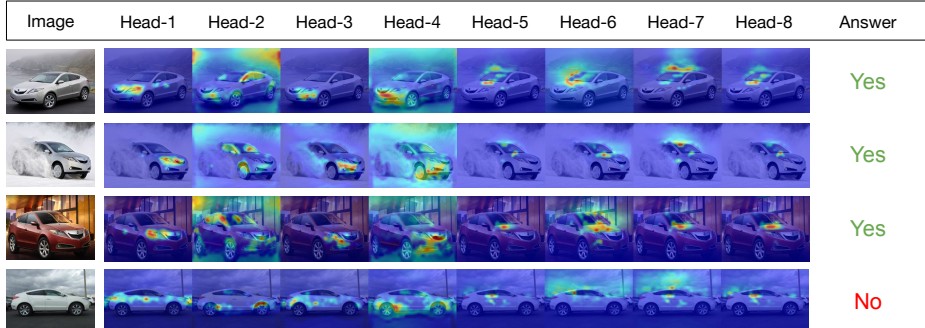

Figure 22: **Illustration of INTR**. We show four images (row-wise) of the same car model Acura ZDX Hatchback and the eight-head cross-attention maps (column-wise) triggered by the query of the ground-truth class. Each head is learned to attend to a different (across columns) but consistent (across rows) semantic cue in the image that is useful to recognize this car model. The exception is the last row, which shows inconsistent attention. Indeed, this is a misclassified case, showcasing how INTR interprets (wrong) predictions.

## H  ADDITIONAL DISCUSSIONS

In this section, we discuss how biologists recognize organisms, how ornithologists (or birders) recognize bird species, and how our algorithm INTR approaches fine-grained classification that could benefit the community.

Biologists use traits — the characteristics of an organism describing its physiology, morphology, health, life history, demographic status, and behavior — as the basic units for understanding ecology and evolution. Traits are determined by genes, the environment, and the interactions among them. Peterson (1999) [6] created the modern bird field guide where he identified key markings to help the birder identify the traits that are distinctive to a species and separate it from other closely related or aligned species. These traits are grouped into four categories: 1) habitat and context, 2) size and morphology, 3) color and pattern, and 4) behavior. Figure 1 shows that INTR can extract the first three categories from the static images, while behavior typically requires videos.

---

[6]This field guide presents the idea of highlighting key features from images to identify species and distinguish them from closely related species. In the field guide, Peterson used expert knowledge to draw a synthetic representation of the species and pointed arrows to the key features that would focus a birder's attention when in the field to a few defining traits that would help the observer to correctly identify it to species.

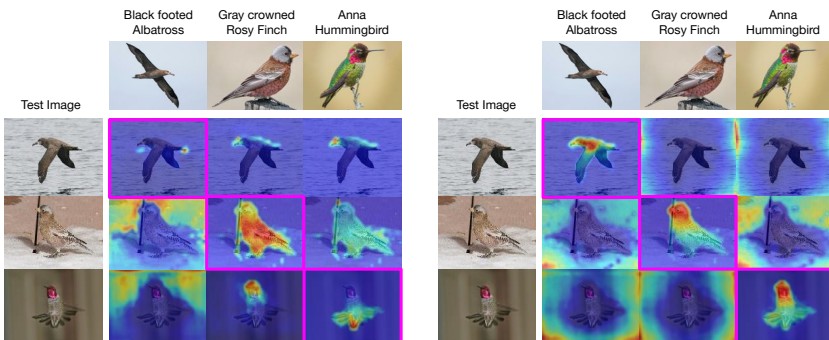

Figure 23: We show the attention maps from two cross-attention heads: Head-1 (left) and Head-4 (right). Row-wise: test images from different classes; column-wise: different class-specific queries. To make correct predictions (diagonal), INTR must find and localize the property of an attribute (e.g., the dotted pattern of the tail from Head-1; the color pattern of the bird's head from Head-4) at the correct part (e.g., on the tail or head).

More specifically, bird field marks that are used by ornithologists often center around two main bird parts: the head and the wing[7]. Patterns of contrasting colors in these regions are often key to distinguishing closely related and therefore similar-looking species. Painted bunting males are nearly impossible to confuse with almost any other species of bunting — the stunning patches of color are highlighted in the results presented in Figure 1. For more dissimilar species, aspects of the shape and overall color pattern are generally more important — as well as habitat and behavior. The characteristic traits of the three species in Figure 23 correspond broadly to these features — the long pointed wings of the albatross; the plump body of the finch; the diminutive feet of the hummingbird. Overall, INTR can capture features at both of these scales, mimicking the process used by humans guided by experts. This is of great potential impact because the analysis of traits is critical for biologists to understand the significance of patterns in the evolutionary history of life. Specifically for fine-grained species, our approach can aid biologists in rapid identification and differentiation between species, refining and updating taxonomic classifications, and gaining insights into relationships between species.

---

[7]Please be referred to (Cor) and (Bir).

