# OpenReview forum: "A Simple Interpretable Transformer for Fine-Grained Image Classification and Analysis"
_ICLR.cc/2024/Conference — ICLR 2024 poster_

### Official Review · Reviewer_8cK8 · 2023-10-24

**Soundness:** 4 excellent
**Presentation:** 4 excellent
**Contribution:** 2 fair
**Rating:** 6
**Confidence:** 5

**Summary:**

This paper presents an interpretable learning framework using Detection Transformer (DETR) for fine-grained classification, namely INTR.

Specifically, learning "class-specific" queries, performing the cross-attention between the queries and the feature maps to obtain "class-specific" features, then a "class-agnostic" "presence" vector to determine where this query (class) is found in the image or not.

This paper also presents a mathematical explanation to unveil how the proposed framework learns.

This paper also presents a few ways to analyze the interpretable classification results, such as image manipulation to monitor changes in cross-attention & the classification score, and fine-grained attributes visualization analysis.

The experiments are mainly performed with fine-grained recognition datasets.

**Strengths:**

originality: While not novel (directly incorporating DETR), this paper rethinks the purpose of DETR and applies it to interpretable learning, also providing a valid mathematical explanation to show why and how the queries learn.

quality: This paper has presented experiments to show the benefit of the proposed interpretable framework. the method is straightforward and is easy to implement.

clarity: The paper is well written. The results are quite clear. But some presentations can be improved, for instance, in fig 5, the author can also show the changes in the attention map (like the one in fig 6). For Fig 4, I am not sure what to look at.

significance: Interpretability is a well-established problem but we still don't have a completely interpretable framework. I think this paper is somehow significant to affect the community. The idea of this paper is simple, this may inspire future works to improve upon this work.

**Weaknesses:**

- Lack of interpretability analysis. As mentioned by the author this paper is not about accuracy, hence I shall expect more quantitative/qualitative experiments to fully understand inner works. Since the author says INTR can detect the "attributes", the author can measure how precise the model can detect. What is the "accuracy" in detecting the attributes (e.g., Sec 4.2 of [1])? What are the captured attributes? Are these attributes shared or different among the different classes (e.g., it can detect the stripe of a bird, are they the same stripes or different in colors/patterns)? Why is that? I think all these results can be shown in the main paper.

- Comparison with other interpretable models. The only comparison I noticed is Fig 3. Are other models not able to identify attributes like INTR does? How do you quantitatively measure it? From what I see in this figure, ProtoPFormer is also able to locate local attributes, but there are not enough samples here.

- Image manipulation experiments seem not systematic enough, how do you measure the changes? E.g., after manipulating the "part", how many images changed the classification results? Are they still the correct or the wrong results?

- The attention map now looks a bit not "clean" enough, when some attributes cannot be detected from the image, I suppose the attention should be empty instead of looking at the background, if it is looking at the background, does it mean it is a feature? or it is overfitting? The author did not further analyze it.

- the sentence "Fortunately, fine-grained classification usually focuses on a small set of visually similar classes; C is usually not large." may be not true, iNaturalist [2] has 10000 classes, and even the "bird" classes alone have 1478 classes.

[1] Zixuan Huang and Yin Li. Interpretable and Accurate Fine-grained Recognition via Region Grouping. CVPR 2020.
[2] https://github.com/visipedia/inat_comp/tree/master/2021

**Questions:**

1. What is the "accuracy" in detecting the attributes (e.g., Sec 4.2 of [1])? What are the captured attributes? Are these attributes shared or different among the different classes (e.g., it can detect the stripe of a bird, are they the same stripes or different in colors/patterns)? Why is that?

2. Are other models not able to identify attributes like INTR does? How do you quantitatively measure it?

3. How do you systematically perform the image manipulation experiments? You should also measure the result quantitatively.

4. I understand that the attention map is not regularized by any loss functions so the attention maps may look weird. But how do you explain the attention focus on the background, but sometimes focus on the "attributes" (e.g. Fig 13)? How can we know what attributes will fire up which heads?

---

> ### Author Response · Authors · 2023-11-23
> **Rebuttal (Part 1)**
>
> **Lack of interpretability analysis.**
> Thank you for the suggestion and reference. We checked the experiment in Sec 4.2 of [1] and found that it may not be feasible for our INTR. First, INTR does not impose regularizers to segment an object into parts. Second, the experiment will require training a regressor to localize part locations, which needs detailed part annotation.
>
> With that being said, we have conducted additional quantitative studies and discussions. First, the attributes that INTR captures are local patterns (specific shape, color, or texture) useful to characterize a species or differentiate between species. These attributes can be shared across species if the species are visually similar. These can be seen in Figure 23 and Figure 24. In Figure 23, we applied the query of Heermann Gull to the image of Heermann Gull (first row) and Ring Billed Gull (second row). Since these two species are visually similar, several attention heads identify similar attributes from both images. However, at Head 8, the attention maps clearly identify the unique attribute of Heermann Gull that is not shown in Ring Billed Gull. Please see the caption for details.
>
> In Figure 24, we present the cross-attention map activated by the ground-truth queries for two closely related species, the Baltimore Oriole and the Orchard Oriole. Additionally, we manually document some of the attributes by checking whether the attention maps align with those human-annotated attributes in the CUB dataset. We found that INTR can identify shared and discriminative attributes in similar classes.
>
> When the species are visually dissimilar (cf. Figure 9), we see that the attributes specifically to each species (every column) cannot be found in other species (the off-diagonal elements of every row).
>
> We will provide more discussions in the final version and move important messages into the main paper.
>
> **Comparison with other interpretable models.**
> Besides Figure 3, we include more comparisons in Figure 10 and Figure 25. As mentioned on page 7 (the paragraph before Sect. 4.2), we do not claim that these compared methods cannot detect attributes. We include them mainly as references to show what our approach can achieve with a simple and extensible model design. One particular advantage of INTR is that it can easily detect attributes of different sizes/shapes. In contrast, ProtoPNet, ProtoTree, and ProtoPFormer often need to pre-define the prototype sizes/shapes.
>
> To quantify the difference between our INTR and ProtoPFormer, we conducted a human study. We randomly selected five images from each of the four species selected randomly from the CUB dataset.  To verify the attributes detected by models, we provide the models’ interpretation attention maps and image-level attribute information (from the CUB metadata) to seven individuals who do not have any knowledge about the project. We ask them to list all the attributes that are captured by the attention maps. One attribute is considered detected by a model if more than half of the individuals identify it from the attention maps. Our quantitative analysis reveals that INTR outperforms ProtoPformer, achieving an attribute identification accuracy of 74.7% compared to ProtoPformer's 42.2%.
>
> We will expand this study to contain more individual observers and bird species in the final version.
>
> **Image manipulation experiments.**
> We initially provided Figure 5 as an analysis, not an experiment. Thanks to your suggestion, and we expanded the study to be more systematic. We considered images from three species of the CUB dataset namely Red-winged Blackbird, Scarlet Tanager, and Orchard Oriole. We removed the red spot of the Red-winged Blackbird (Figure 5), edited Scarlet Tanager by changing the black wings and tail to red to make it look like Summer Tanager (Figure 26), and replaced the color of Orchard Oriole’s belly from brown to orange to make it look like Baltimore Oriole (Figure 5). All of these manipulations follow the attribute guidance described in https://www.birds.cornell.edu/home/. We randomly selected 10 images from each of these species for manipulation and observed that 29 cases changed their prediction after manipulation (out of 30), which shows a 96.7% success rate. This holds the property that INTR can identify tiny image manipulations that distinguish between classes.

---

> ### Author Response · Authors · 2023-11-23
> **Rebuttal (Part 2)**
>
> **Attention maps.**
> Thank you for the comments. It is worth noting that in our original submissions, Figures 12-17 are indeed hard classes, as we want to ensure that every picked class has misclassified examples (the last row) to match the presentation style of Figure 1. Given the high performance of the Butterfly (BF) dataset, selecting a challenging case (the one pointed out by the reviewer in Fig. 13) was necessary to maintain this pattern.
>
> In the rebuttal, we further present Figures 27-28, which are classes without misclassified cases. These attention maps are much clearer.
>
> Furthermore, it is worth highlighting that the attention heads presented are normalized based on attention scores, which may, in some instances, result in the background being highlighted. However, when we sort them based on the un-normalized scores, we find that only the essential attributes are highlighted, as illustrated in Fig. 4.
>
> We also note that, according to https://www.birds.cornell.edu/home/, habitats are also considered important messages to differentiate species. Thus, we do expect for some species, the attention maps may focus on the background.
>
> **About the statement in the limitation.**
> We apologize if we overclaim. We mainly want to mention that this is a usual case. For all the fine-grained datasets we consider, they are within 1000 categories. We will certainly clarify this.
>
> We appreciate that you point out the case of bird species in iNaturalist. Please note that, while the whole iNaturalist contains >10K species, those species span across multiple Kingdoms, Phylums, etc. We thus do not consider inputting all the 10K classes all at once into INTR, but focus on recognizing species of the same Class (e.g., Class Animalia Chordata Actinopterygii) or Order. We will certainly clarify this.
>
> *In light of our clarification and additional results, we would like to ask if you are willing to reconsider your score. Thank you.*

---

### Official Review · Reviewer_Z5oh · 2023-10-31

**Soundness:** 3 good
**Presentation:** 4 excellent
**Contribution:** 3 good
**Rating:** 6
**Confidence:** 4

**Summary:**

The authors present a method that uses Cross-Attention in a Transformer Encoder-Decoder setup to provide interpretable insights how classification results are obtained. By learning class-specific queries as input to the decoder paired with a shared class-agnostic weight vector, the method checks in an intuitive way whether each class can be located in an image, and which attributes within the image lead the ‘classifier’ to believe a certain class is present/dominant – providing a faithful way of interpreting how the decision was made (via visualization of the respective attention patterns).

**Strengths:**

### Quality:
- Method is described in sufficient detail, including the intuitions underlying each component; Authors additionally do a good job in providing enough mathematical background to clearly follow the inner workings in a concise way;
- Conducted experiments clearly outline and support the intended contributions, mostly via illustrations of ‘qualitative’ results
- The level of detail provided in Section 3.4 is very helpful to further understand underlying motivations and inner workings of the method;
### Clarity:
- Very easy to read and follow, very well-structured manuscript; The author’s do an excellent job in posing questions that are then answered to clearly articulate their goals; Good use of illustrations to create compelling story line
### Originality & Significance:
'Originality' is (as often) slightly tricky here: The underlying idea of using cross-attention between learnt queries that try to locate itself in the image is pretty much directly adopted from DETR – as the authors themselves correctly state.
&#8594; The originality of this work hence mainly lies in the application to use this technique for better interpretability, which is paired with/supported by their way of performing classification to support faithfulness of the involved attention patterns;
Considering these points, I do support the author’s claim of originality in this sense, especially thanks to the explicit consideration of the faithfulness aspect.

**Weaknesses:**

**TLDR:** While one could potentially criticize the novelty in terms of methods, I do think this work clearly outlines a different and interesting perspective how CA can be used to increase interpretability in a (more) faithful way. Paired with the well-structured manner the manuscript is prepared, I lean towards recommending acceptance of this work — and am happy to further increase my score if the authors can address my questions & concern (mainly 1st weakness).

---
*Missing discussion of the encoder’s potential influence*:
The (potential) influence of the encoder is not really discussed in this work; Standard DETR already employs a quite large CNN to ‘encode’ the information into embeddings, and one could argue that certain ‘decisions’ as to what is important are already made there. Note that the receptive field of each element of the feature map emitted by the encoder is substantial (potentially even global, esp. when ViT is used) and therefore could incorporate information of various locations throughout the actual input image.
&#8594; Please correct me if I’m wrong, but the CA essentially shows us to which locations of the feature map the classifier is paying attention to – which does not necessarily have to directly correspond to the input image? I’d like to see some discussion/clarification of this point.

---
*Known and stated weaknesses/limitations, but still worth pointing out*:
-  Self-attention in the Transformer Decoder quickly becomes prohibitive – as the authors correctly state. It however still worth noting that the complexity in memory and time grows quadratically with the number of classes due to the self-attention operation performed across the class-specific query tokens.

    &#8594; I’d be curious to know whether the authors might have some suggestion whether all classes are required to be included as decoder input ‘all the time’, or whether some pre-selection could be performed to reduce the computational complexity, at least at inference time.


- Influence of pre-training: The authors state that they employ the DETR model pretrained on ImageNet and MSCOCO – which is a lot of data that importantly provides various different aspects of images (classification, as well as (multi-object) detection data);
Note: The authors acknowledge this fact in the manuscript, but it is still worth pointing out since it is a potential point of concern that’s introducing some uncertainty as to how backbones that are trained on less data or ‘only ImageNet’ might perform when paired with INTR.

**Questions:**

**Main Questions connected to previously mentioned 'weaknesses'**:

- Related to **Influence of Encoder**: As previously detailed, the CA shows to which locations of the feature map the classifier is paying attention to – which does not necessarily have to directly correspond to the input image, especially for architectures with global receptive fields. Do the authors agree with this? What are the (potential) implications of this towards the interpretability of the presented approach?

- Related to **Efficiency**: Do the authors see any possibility to improve efficiency of the method in terms of preventing quadratic growth w.r.t. the number of classes? (and thus increasing the ease of applicability towards datasets with more classes)

- Related to **Pre-training influence**: Did the authors experiment with training from scratch, e.g. on ImageNet, as well? (One alternative could also be using a (frozen) pre-trained backbone and train the 'remainder' of the architecture (Transformer decoder) on only ImageNet); Do the authors have any insight as to how much training on a detection task might help with focusing on detail, and whether this makes any difference to 'simple' classification pre-training?


---
Some additional questions (mainly out of curiosity):

- I’d like to get some more insight regarding the importance of the shared ‘w’ vector to perform the classification; While I do understand the intuition, would the result of learning distinct representations differ if the representation itself was just averaged, i.e. ‘w’ would simply be 1/len(w)?

- Simply out of interest: Did the authors encounter images with multiple objects during their experiments? If so, was their method able to actually locate these individually within the image if queries for the (multiple) ‘correct’ classes existed? If not, how would the authors expect the method to behave – ‘correctly’ finding e.g. 2 objects in an image, or do the authors expect that such use cases will cause issues? (Such a case could potentially be constructed by simply concatenating 2 partial images)

---
Comment:
- It might be worth changing the way how HxW is described: The authors consistently describe ‘H’ and ‘W’ as the ‘number of grids’ – however, I’d strongly suggest to instead refer to the elements as “grid cells”, since at least I would see the entire image/matrix HxW as the grid, with H and W indexing the ‘grid cells/elements of the grid’


-----
**Post rebuttal update:**
I'd like to genuinely thank the authors for their detailed answers and the effort invested into this rebuttal!
My main concerns regarding the encoder's influence (receptive field) and pre-training have been partially addressed;
I do however see some remaining challenges in the presented insights regarding reduced interpretability when using a different backbone: While I agree with the authors' statement that reuse of pretrained backbones is legitimate (and should be encouraged), the results show that using another (non-DETR) pretrained backbone (e.g. the presented ViTs in Fig.22) significantly hinders the efficacy of the presented INTR; This might be due to misalignment, but note that if the method should be useful, it must be applicable to a variety of backbones of interest, not just to one specific combination that has been optimized for DETR; Whether the method transfers well onto other architectures of choice still remains somewhat open/not entirely satisfyingly resolved.
$\rightarrow$ That said, I do think the paper presents a novel and insightful take on interpretability. After reading the other reviews, I decided to stick with my initial rating.

---

> ### Author Response · Authors · 2023-11-23
> **Rebuttal (Part 1)**
>
> **Influence of Encoder.**
> Thank you for the insightful question. The large receptive field of a deep CNN or a plain Transformer indeed indicates that every feature map location has been affected by many pixel locations in the input image. On the one hand, this helps capture the context in the image (to differentiate visually similar pixels or local patterns). On the other hand, it may lose the localizability — each feature map location does not necessarily correspond to its associated pixel/patch locations.
>
> That said, we think several design choices in ResNets and the Transformers used in vision (e.g., ViT or the encoder in DETR) help improve localization. First, both of them have “skip” links. That is, there is a shortcut from the input pixel/patch to the corresponding feature map location. Such a skip link has been shown to improve localization in semantic segmentation (e.g., U-Net) and object detection (e.g., spatial pyramid pooling) as well. In [A], the authors empirically showed that a significant amount of feature information is passed from the previous layers to the later ones through the skip links. Second, in the encoder of DETR, fixed positional encodings are added to all the encoder layers. We surmise that this would further strengthen the location correspondence between the input image and the feature maps. Finally, as evidenced by recent work of ViTs like DINO-v1 and DINO-v2, we do see a fairly accurate correspondence of the object/part boundaries in the feature map and in the image.
>
> We hope that these discussions address your concern about the interpretability of INTR.
>
> [A] Maithra Raghu et al., Do Vision Transformers See Like Convolutional Neural Networks? NeurIPS 2021
>
> **Efficiency.**
> Thank you for the insight on the quadratic growth of computation. We have indeed been investigating methods to reduce it. The first method is to perform hierarchical classification following the available taxonomy (e.g., the organism taxonomy). We can learn queries for different taxonomy levels (e.g., for different families, genera, and species). During inference, we then apply these sets of queries in sequence. In other words, we do not need to input all the species-level queries to the decoder at once; only those belonging to the predicted genus are needed. Consider a two-level hierarchy with C^2 classes, where every C of them is grouped. Hierarchical inference takes 2*O(C^2) computation, largely saving it from O(C^4) by the plain inference.
>
> Even without hierarchical training, one may take a pre-trained INTR, perform hierarchical clustering to group the classes (e.g., based on the validation confusion matrix), and select one representative class to represent each cluster. During inference, we then use those representative classes, from the root of the hierarchy down, to perform hierarchical classification. In Fig. 21, we show an example that the top predicted classes by INTR indeed share similar appearances and genera; one can thus use one query to represent the others. In Fig. 6, we also show that, by taking only the most visually similar queries as input, INTR can identify finer-grained attributes.

---

> ### Author Response · Authors · 2023-11-23
> **Rebuttal (Part 2)**
>
> **Pre-training influence.**
> Thanks for carefully checking our paper, especially the footnote on page 6 that discusses our usage of an MSCOCO-pre-trained model.
>
> During the rebuttal period, we tried to pre-train the INTR model from scratch, i.e., randomly initializing the Transformer encoder and decoder in DETR, using the ImageNet data. Unfortunately, we found the hyper-parameter setting of DETR not directly applicable to this pre-training. While we have tried to perform hyper-parameter selection, we were short of time to find a good configuration. We will try to continue the study.
>
> As an alternative, we take the pre-trained DETR model and replace its encoder (i.e., ResNet-50 + Transformer encoder) with a separately pre-trained DeiT-S (using ImageNet-1K) and ViT-H (using ImageNet-21K). Details of this study are in Appendix Section F. We found that using a DeiT-S or ViT-H encoder trained without MSCOCO can sometimes outperform the original ResNet-50 + Transformer encoder in terms of **accuracy**, as shown in Table 4. However, in terms of the  **interpretation/visualization**, using the pre-trained DETR model is still the best. Figure 22 shows an example. We attribute this to two reasons: 1) the pre-trained DeiT-S and ViT-H are not well-aligned with the pre-trained decoder, and 2) pre-training with detection data does help with focusing on details.
>
> With that being said, we respectfully think that these results should not undermine our main contributions and claims. First, DETR is a model architecture designed initially for detection. Training it from scratch may need some careful investigation. Second, with the accessibility to large-scale detection datasets and pre-trained models, building a model from the most suitable starting point should be allowed and encouraged. In our case, the MSCOCO dataset does not contain fine-grained class labels and part annotations, quite different from our downstream tasks. Besides, what we found interesting is that while existing literature mostly uses classification pre-trained models to facilitate other downstream tasks, we show that the opposite—using detection pre-trained models to facilitate classification—is also worth exploring.
>
> **The shared w vector.**
> The “w” vector is a shared binary classifier telling whether a class is seen in the image. During training, it is optimized to align with the outputted vector $z_{\text{out}}^{(c)}$ of the true class (cf. Sect. 3.4). In theory, if the model’s capability is large enough, one could freeze “w” (e.g., 1/len(w), as suggested by the reviewer) and learn the model backbone to generate $z_{\text{out}}^{(c)}$ that aligns with the frozen “w”. As all the classes still share a single “w”, the argument of learning distinct queries per class should still hold (cf. Sect. 3.4).
>
> In practice, we found that freezing “w” makes the training much harder. For example, using the same learning rate and total epochs, the trained model can only obtain a 76.3 classification accuracy on the Pet dataset, compared to 90.4 w/o freezing.
>
> **Images with multiple objects.**
> This is an interesting question. As we mainly work on classification datasets, we expect that most images only contain one object (or objects of a single class). We follow your suggestion to stitch pictures of two bird species. We then check the cross-attention maps triggered by each of the two queries. We have the following observations.
>
> 1. The two sets of cross-attention maps (triggered by the two queries) can jointly localize both object instances.
>
> 2. Each query focuses a bit on the corresponding object instance.
>
> 3. Several cross-attention maps distribute their weights to both object instances, especially when the two bird species share some common attributes (e.g., head shapes).
>
> We attribute the observation in point 3 to 1) the mismatch between training and testing and 2) the inner workings of INTR — it aims to find class-specific patterns but may not localize them from the same object instance. In practice, one may use INTR with an object detector, using the object detector to identify object instances (e.g., birds) and then applying INTR to each separately.
>
> Regarding the classification result on the stitched image, we found that INTR tends to rank a third class that contains the attributes of both stitched classes higher than each of them. For example, INTR predicts “Magnolia Warbler” on the stitched images of “Green Violetear” and “Prairie Warbler”; INTR predicts “Painted Bunting” on the stitched images of “Indigo Bunting” and “Summer Tanager”. This result makes sense regarding how INTR was trained: it was trained with the cross-entropy loss to output the most likely class by checking whether its attributes are found in the image.
>
> **HxW.** Thank you for the suggestion, and we will use grid cells in our final version.
>
> *In light of our clarification and additional results, we would like to ask if you are willing to reconsider your score. Thank you.*

---

### Official Review · Reviewer_dzqa · 2023-11-01

**Soundness:** 3 good
**Presentation:** 3 good
**Contribution:** 2 fair
**Rating:** 6
**Confidence:** 4

**Summary:**

This paper introduces a method called feature accentuation. It is a new explainability method that it indicates which pixels in the image are relevant for the final decision of the model. As well as which kind of features activate relevant neurons.

**Strengths:**

A strength of this method is not needed auxiliary generative models and being seeded to images.
Also, the release as an open-source library as part of Lucent will make it accessible to those who would like to use it for their applications or built on top of it.

To overcome some gaps from previous related research, this methods incorporates several techniques  to avoid that the modified image that accentuates some feature changes how it activates the neuron’s with respect to the original seed image as a regularisation term in the loss. There is an analysis on which layers play an important role to avoid undesired distortions, and it is reported that enforcing it in earlier layers yields better visualizations in early layers. An analysis in the impact of regularisation, parametrisation and augmentation techniques from the literature applied in this method is conducted, highlighting the right combination of those factors. Additionally, to improve relevance of the feature representations, a global normalisation is proposed

The experiments reported use circuit coherence assessment from another paper, which is a reasonable measure.

**Weaknesses:**

The other applications showcased are also of high importance, but the results become more difficult to assessed. How confirmation biased is overcomed with this method? It is still based on visualisations which need human assessment. The what is based on visual information that is difficult to parse for a human. How useful then it really is still an open question. This is already mentioned in limitations, but it is a strong self-critic that should be given more thought on how to overcome those. How to in corporate over tools for the interpretation is also not clear.



MINOR:
There is a reference missing with a question mark. There are a couple of blank space missing to segment words.

**Questions:**

Please refer to weakness points.

**Details Of Ethics Concerns:**

It's a minor concern. It is not clear how useful is the explainability with this tool.

---

> ### Author Response · Authors · 2023-11-11
> **Sincere request for clarification**
>
> Dear Reviewer dzqa,
>
> We appreciate your time and effort in reviewing our paper. We are glad to see your positive rating.
>
> However, after a careful reading of your feedback, we found that it may not directly relate to our paper content. May we kindly request that you take a look at this potential problem? Thank you.
>
> Best,
> Authors

---

> > ### Comment · Area_Chair_i3N6 · 2023-11-11
> > **RE:Sincere request for clarification**
> >
> > Thank you for reaching out, we are looking at it.
> > Let us come back to your shortly.
> >
> > regards

---

> > ### Comment · Reviewer_dzqa · 2023-11-15
> > **review content correction**
> >
> > Dear authors,
> >
> > thanks for reaching out and pointing out at the mismatch. Indeed, the content of the review is not about your submitted paper. I apologize for my confusion.
> >
> > Here I post a revised feedback.
> >
> > This paper proposes to add the class at the input of the enconder of an ecoder-decoder transformer model and claim that this leads to an interpretable classifier based on transformer architecture that is suited for interpretable fine-grained cassification.
> >
> > The simplicity of the the method is the main strength of the paper. It reports results in a plethora of datasets to showcase the soundness of the method, highlighting which specific features are relevant for the classification task.
> >
> > In the results section, when analysing the performance of the proposed model, the comparision to the ResNet model as baseline is not completely justified in this set up. I believe that the comparison should be with the same architecture without the class query at the decoder and other similar architectures changing the relevant parameters.
> >
> > On another more high-level argumentation of importance of the proposed model, the relevance of interpretability that is reported is arguable because it is not quantified with humans in the loop. It is also not clear how different this is with the typical existent transformer architectures. A more meaningful comparison would strengthen the paper contribution.
> >
> > Nevertheless, this line of research is indeed relevant, and when adding further discussions, and if possible more baselines for comparison, I foresee that this paper can influence future research to mitigate the mentioned limitations.

---

> > > ### Author Response · Authors · 2023-11-15
> > > **Re: review content correction**
> > >
> > > Dear reviewer,
> > >
> > > We appreciate your quick reply with the revised feedback. We wondered whether you could update your review in the original form so we can get a holistic picture of the review, including Soundness, Presentation, Contribution, Rating, Confidence, Flag For Ethics, and Details Of Ethics Concerns.
> > >
> > > Thank you so much.
> > >
> > > Best,
> > > Authors

---

> > > ### Comment · Area_Chair_i3N6 · 2023-11-16
> > > **RE: review content correction**
> > >
> > > Please share the
> > >
> > > Questions:
> > >
> > > Flag For Ethics Review:
> > >
> > > Details Of Ethics Concerns:
> > >
> > > Rating:
> > >
> > > Confidence:
> > >
> > > Code Of Conduct:
> > >
> > >
> > > Thank you
> > >
> > > Regards

---

> > > > ### Comment · Reviewer_dzqa · 2023-11-16
> > > >
> > > > Questions:
> > > > What are the results compared to other models rather than ResNet? In particular, with the same architecture without the class query at the decoder and other similar architectures changing the relevant parameters.
> > > >
> > > > Comment on the relevance of interpretability that is reported. It should be quantified systematically, for instance with humans in the loop.
> > > >
> > > > Flag For Ethics Review: Yes, others
> > > >
> > > > Details Of Ethics Concerns: It's a minor concern. It is not clear how useful is the explainability with this tool.
> > > >
> > > > Rating:
> > > >
> > > > Confidence:
> > > >
> > > > Code Of Conduct:
> > > >
> > > > Rating: 6: marginally above the acceptance threshold
> > > > Confidence: 4: You are confident in your assessment, but not absolutely certain. It is unlikely, but not impossible, that you did not understand some parts of the submission or that you are unfamiliar with some pieces of related work.
> > > > Code Of Conduct: Yes

---

> ### Author Response · Authors · 2023-11-23
> **Rebuttal**
>
> **Comparisons to models other than ResNet.** Thank you for the comment and suggestion. We consider two variants of INTR. **1) INTR-FC:** the same architecture as INTR except for learning a class-specific $w_c$ at the model’s output (cf. Sect. B and Eq. 10). This baseline was included in our original submission, mainly to demonstrate the effectiveness of INTR’s classification rule on interpretability: INTR incorporates class information at the decoder’s input (cf. Eq 2 & 6; the last paragraph of Sect. 1) but not at the model’s output (cf. Eq. 1 & 10). With additional model capacity, INTR-FC achieves a higher accuracy (77.5 and 92.7 on CUB and Pet, respectively) than INTR (71.8 and 90.4). Nevertheless, as shown in Fig. 7, INTR-FC obtains a worse interpretation than INTR, as it does not hold the nice properties of INTR that encourage interpretability (cf. Sect. 3.4 and Sect. B). **2) INTR-Enc:** We remove the decoder of INTR and add a fully-connected layer on top of the encoder’s output. (This is to implement the reviewer’s suggestion. We respectfully think there is no need to have the decoder if the class-specific queries are disregarded.) With the fully-connected layer on top, INTR-Enc achieves slightly higher accuracy (73.0 and 90.7 on CUB and Pet, respectively) than INTR but loses the interpretability of INTR.
>
>
> We hope that the results above support our statements in the main paper (the paragraph before Sect. 4.1 and the first paragraph of Sect. 4.1): *achieving a high classification accuracy is not the goal of this paper.; the goal is to demonstrate the interpretability.* In Table 2, we compared INTR to ResNet-50 mainly because it is a building block in INTR. Nevertheless, the results are mainly as references, not to claim that INTR outperforms ResNet-50 in accuracy, and we will clarify this.
>
>
> **Quantitative interpretability results. Humans in the loop.**
> Thank you for the suggestion. During the rebuttal, we conducted a human study comparing INTR and ProtoPFormer. Please kindly refer to the second response to reviewer 8cK8. We also provided more systematic results on the manipulation analysis in Figure 5. Please see the third response to reviewer 8cK8. We also offered results using metrics in XAI. Please see the second response to reviewer 4rTr.
>
> We will include the above results in the final version and seek other quantitative metrics to strengthen our paper. To our knowledge, besides showing classification accuracy, prior work on interpretable methods (e.g., ProtoPNet, ProtoPFormer, and ProtoTree) has mostly reported qualitative results. Compared to them, we have conducted more qualitative analyses and provided more visualization results to showcase the inner workings of our approach.
>
> **Comparison to typical transformer architectures in terms of interpretability.** Thanks for the question. We note that typical transformer architectures for classification (e.g., ViT or Swin-Transformer) follow the classification rule in Eq (1). Thus, a post-hoc explanation method like Grad-CAM is needed to explain why the model predicts a particular class. Even though one may analyze the attention weights within the model, they may fall into the debate mentioned in Sect. 3.4. We also kindly refer the review to Sect. 3.5, where we provide further discussions between INTR and closely related work.
>
>
> **Ethics concerns.** Our method aims to interpret the model’s prediction, which can potentially improve the trustworthiness and discover new knowledge for prediction (e.g., new attributes for animal species). We humbly think that our approach does not introduce any additional ethical concern or negative societal impact than other explainable and interpretable work.
>
> *In light of our clarification and additional results, we would like to ask if you are willing to reconsider your score. Thank you.*

---

### Official Review · Reviewer_4rTr · 2023-11-04

**Soundness:** 3 good
**Presentation:** 4 excellent
**Contribution:** 2 fair
**Rating:** 6
**Confidence:** 4

**Summary:**

The paper introduces Interpretable Transformer (INTR), a classifier that builds on the standard Transformer architecture. It leverages the cross-attention and a set of learnable class-specific queries to introduce by-design visual explanations for each class. The interpretability of INTR comes from the ability to track the cross-attention weights during inference, showing which parts of an image are being considered when making a prediction. The model proves to be effective in not only identifying parts of objects like bird heads but also in distinguishing between species by recognizing subtle attributes. The decrease in classification accuracy compared to ResNet is not a big deal but the evaluation that is mostly qualitative makes me question about its utility. Found details below.

**Strengths:**

- The presentation is clear and structure is easy-to-follow.
- I like the idea of manipulating distinctive features in Figure 5 and believe the authors could relates to Causal inference.
- I like the honesty in Table. 2 where the authors show the bad performance of INTR over different datasets.

**Weaknesses:**

- As this paper’s evaluation contains mostly qualitative results (e.g. showing the visualizations and explanations of INTR), it would be great if they can show the real utility of the explanations for a human-related task [a]. In this case, the decreases in accuracy shown in Table. 2 can be completely negligible. Otherwise, I still see ResNet clearly better than INTR (although this is not a major point).
- The authors may also want to evaluate the explanations using proxy metrics in XAI.
- It has been unclear to me whether the bad performance of INTR stems from the capability of query vectors or not? It would be great if the authors have ablation studies for the query vectors (e.g. for classes with high inter-class variations, a simple vector could not be representative for the whole class). Also, how do they affect the explanations.
- The number of heads is equivalent to the number of concepts in ProtoPNet and its variants. For CUB, I believe the number of distinctive features is less than 8 (to my experience). Then I believe this number of heads should be adjusted according to the dataset.

[a] visual correspondence-based explanations improve ai robustness and human-ai team accuracy

**Questions:**

N/A

Post-rebuttal reviews:

I genuinely appreciate your great efforts put into this rebuttal!
# Real utility of the explanations.
Authors insisted that:

> In our current paper, the targeted real utility is indeed automatic attribute/trait identification and discovery for organisms.

and in Sect. G, they provided more context:

> These traits are grouped into four categories: 1) habitat and context, 2) size and
morphology, 3) color and pattern, and 4) behavior. Figure 1 shows that INTR can extract the first
three categories from the static images, while behavior typically requires videos.

However, looking at Fig.1, it is unclear how INTR extracts size and morphology information. For added context and background, simply highlighting background pixels provides only loose support for INTR's ability to extract habitat-related traits. There is a need for a systematic evaluation of INTR's efficiency in trait discovery. Currently, most of the claims about its utility are subjective and unconvincing..

# Evaluate the explanations using proxy metrics in XAI.

As authors also stated that:
> For self-interpretable methods like ProtoPNet, ProtoPFormer, ProtoTree, and our INTR, which build specific classifiers, XAI metrics are seldom reported as they may not be fair across different methods.

The current evaluation using deletion and insertion as proxy metrics appears to be of limited significance. A more meaningful benchmark would involve comparing the explanations with other concept-based, self-interpretable classifiers, rather than relying on post-hoc explainers.

# Performance of INTR.

Does increasing M to 2 equate to doubling the number of heads, in terms of the concepts to be learned? In Fig.20's bottom row, while increasing M can rectify misclassifications, the resultant explanations tend to be noisier. I hope authors will revise this for the next version.

# Overall

I again thank the authors for the incredible efforts in the rebuttal. Given that the authors partially addressed my concerns, I will increase the score from 5->6.

---

> ### Author Response · Authors · 2023-11-23
> **Rebuttal (Part 1)**
>
> **Real utility of the explanations.**
> We appreciate your feedback. To our knowledge, in many scientific or safety-critical domains, such as biodiversity and medical applications, domain experts often care more about whether the model is trustworthy or can discover new knowledge (e.g., traits for animals or biomarkers for diseases) rather than purely achieving higher accuracy. In our current paper, the targeted real utility is indeed automatic attribute/trait identification and discovery for organisms. This is of great impact because the analysis of traits, the integrated products of genes and environment, is critical for biologists to predict the effects of environmental change or genetic manipulation and to understand the significance of patterns in the evolutionary history of life. Specifically for fine-grained species, our approach can aid biologists in rapid identification and differentiation between species, refining and updating taxonomic classifications, and gaining insights into relationships between species.
>
> To demonstrate this capability, we conduct additional analyses focusing on closely related pairs of butterfly species within the Cambridge Butterfly dataset. Our objective was to pinpoint the distinctions discernible by our model and compare them with the differences identified by experts. Specifically, we selected three finer species (Heliconius melpomene and Heliconius elevatus; Heliconius erato and Heliconius melpomene; Heliconius melpomene and Heliconius erato) from the Cambridge Butterfly website and confirmed that our method effectively detects these differences as shown in Figure 18 and 19 (and Figure 6). This implies that our approach can assist experts in accurately discerning (and potentially discovering) the unique characteristics of fine-grained species.
>
> The traits identified by human experts can be found here.
>
> Website: https://www.cliniquevetodax.com/Heliconius/index.html
>
> Figure 6: https://www.cliniquevetodax.com/Heliconius/pages/melpo%20vs%20elevatus.html
>
> Figure 18: https://www.cliniquevetodax.com/Heliconius/pages/melpo%2520vs%2520erato.html
>
> Figure 19: https://www.cliniquevetodax.com/Heliconius/pages/erato%20favorinus.html
>
> Finally, as discussed in the first response to reviewer dzqa, improving the classification accuracy may hurt the model’s interpretability. We thus think it is fine to lose the classification accuracy a bit for trading better interpretability.
>
> **Evaluate the explanations using proxy metrics in XAI.**
> Thank you for the suggestion. In our original submission, we indeed included results using proxy metrics in XAI (cf. Sect. F and Table 6 in the Appendix). We apologize if we did not make it clear.
>
> Specifically, we applied the *deletion* and *insertion* metrics. Given a post-hoc explanation method that generates a saliency map for the input image (e.g., Grad-CAM, cf. Sect. 2.2 and Sect. A), these metrics first sort the image patches based on the saliency scores (descending order). The deletion metric then iteratively removes patches from the image, inputs the manipulated image to the classifier, and measures the confidence of the correct class. The more sharply the confidence drops (i.e., the lower the AUC is), the better the explanation is, as it highlights the most discriminative patches, which, once removed, would drastically reduce the classifier’s confidence. In contrast, the insertion metric blurs the original image and iteratively discloses the patches. The more sharply the confidence increases (i.e., the higher the AUC is), the better the explanation is, as it highlights the most discriminative patches, which, once inserted, would drastically improve the classifier’s confidence. Please see more details in Sect. F, paragraph: Comparisons to post-hoc explanation methods.
>
> The results are shown in Table 6 of the Appendix (during the rebuttal, we further include the results of RISE). INTR notably outperforms Grad-CAM and RISE, achieving a higher insertion AUC and a lower detection AUC.
>
> That said, XAI metrics are designed mainly for post-hoc explanation methods like Grad-CAM and RISE, which aim to explain a black-boxed classifier like ResNet. For self-interpretable methods like ProtoPNet, ProtoPFormer, ProtoTree, and our INTR, which build specific classifiers, XAI metrics are seldom reported as they may not be fair across different methods.

---

> ### Author Response · Authors · 2023-11-23
> **Rebuttal (Part 2)**
>
> **Performance of INTR.**
> Thank you for the insightful comment. In analyzing INTR’s prediction, we did find that some errors result from intra-class variations. Specifically, for many bird species, different genders and life stages show different appearances and attributes (https://www.allaboutbirds.org/guide/Painted_Bunting). For example, in Fig. 1, the last row (misclassified) is indeed a female Painted Bunting; others, male Painted Bunting. These intra-class variations can be considered *sub-categories*, and we surmise that learning one query vector per class may only capture the major sub-categories in the dataset but not the entire variations.
>
> During the rebuttal period, we investigated the idea of learning *M* queries per class. This idea results in M logits per class (cf. Figure 2), and we averagely pool them to represent each class in both training (for the cross-entropy loss) and in testing. (We also investigated max pooling but found it hard to converge.) We see a notable gain on the CUB dataset: using $M=2$ queries per class improves INTR from **71.8** to **75.7**. We also include a qualitative result in Fig. 20. The image is an immature male Hooded Oriole, and INTR misclassifies it. Using $M=2$ queries, the model can correctly classify the image and highlight the black throat attribute. We will include a comprehensive evaluation of this multi-query version of INTR in the final version.
>
> In our humble opinion, the improvement by using multiple queries per class does not change our main paper's main contributions and messages. In contrast, it further showcases INTR’s extendability and its potential to discover sub-categories within a class.
>
> **The number of heads.**
> Thank you for the comment. In our humble opinion, the number of heads in INTR can be different from the number of concepts per class in ProtoPNet. In ProtoPNet, each class is allowed to learn a separate set of prototypes; in INTR, the model parameters of the cross-attention heads are shared by all classes. About the number of discriminative features, according to the CUB dataset, there are 312 human-annotated attributes across more than 20 part-{pattern, shape, size} combinations. Thus, we respectfully think it is reasonable for INTR to use eight or more heads.
>
> That said, the number of heads is adjustable as it is essentially a hyper-parameter. In our original submission, we mainly followed the DETR paper to use eight heads. We did conduct an ablation study on the CUB dataset in Table 5 and found that eight heads slightly outperform other numbers. During the rebuttal, we further compare different numbers of heads for the Butterfly and Pet datasets and see similar trends. We will include a more comprehensive ablation study in the final version.
>
> *In light of our clarification and additional results, we would like to ask if you are willing to reconsider your score. Thank you.*

---

### Meta-Review · Area_Chair_i3N6 · 2023-12-05

**Metareview:**

Dear authors,

This is an interesting problem, and reviewers have assigned a ranking 6 (marginally above the acceptance threshold), with one reviewer increasing the score from 5 to 6 after rebuttal.

We encourage authors to update the draft by incorporating suggestions.

regards

Meta reviewer

**Justification For Why Not Higher Score:**

All reviewers have assigned rank 6 and not more.

**Justification For Why Not Lower Score:**

All reviewers have assigned rank 6, its an interesting problem and solution seems reasonable.

---

### Decision · Program_Chairs · 2024-01-16

Accept (poster)